# EVALUATING TEXT CREATIVITY ACROSS DIVERSE DOMAINS: A DATASET AND LARGE LANGUAGE MODEL EVALUATOR

**Qian Cao**[1], **Xiting Wang**[1✉], **Yuzhuo Yuan**[2], **Yahui Liu**[3], **Fang Luo**[2], **Ruihua Song**[1✉]
[1]Renmin University of China, [2]Beijing Normal University, [3]Kuaishou Technology
{caoqian4real, xitingwang, rsong}@ruc.edu.cn, joyyuan@mail.bnu.edu.cn,
yahui.cvrs@gmail.com, luof@bnu.edu.cn

⌂ Project Page: https://creval-creative-evaluation.github.io

## ABSTRACT

Creativity evaluation remains a challenging frontier for large language models (LLMs). Current evaluations heavily rely on inefficient and costly human judgments, hindering progress in enhancing machine creativity. While automated methods exist, ranging from psychological testing to heuristic- or prompting-based approaches, they often lack generalizability or alignment with human judgment. To address these issues, we propose a novel pairwise-comparison framework for assessing textual creativity that leverages shared contextual instructions to improve evaluation consistency. We introduce CreataSet, a large-scale dataset with 100K+ human-level and 1M+ synthetic creative instruction-response pairs spanning diverse open-domain tasks. Through training on CreataSet, we develop an LLM-based evaluator named CrEval. CrEval demonstrates remarkable superiority over existing methods in alignment with human judgments. Experimental results underscore the indispensable significance of integrating both human and synthetic data to train highly robust evaluators, and showcase the practical utility of CrEval in boosting the creativity of LLMs.

## 1 INTRODUCTION

Creativity, defined as "ideas or artifacts that are new, surprising and valuable" Boden (2003), has long been a defining trait of human intelligence and fueled the progress of modern civilization Guilford (1967). As current large language models (LLMs) exhibit increasingly remarkable capabilities across diverse domains and downstream tasks, they have also shown the ability to perform tasks requiring creativity Summers-Stay et al. (2023); Zhao et al. (2025); Zhong et al. (2024); Wu et al. (2025b). Evaluating the creativity of LLMs not only sheds light on their applicability to critical creative domains such as creative writing Chakrabarty et al. (2025); Marco et al. (2024), literature Bena & Kalita (2019); Cao et al. (2022); He et al. (2023) and other creative domains Naeini et al. (2023); Summers-Stay et al. (2023); Tian et al. (2024), but also has the potential to reveal gaps between LLM and human capabilities, offering valuable insights for future improvements.

Although evaluating LLM creativity is increasingly important, current methods face limitations that restrict their broader applicability. First (**cross-domain applicability**), most current methods assess creativity within a single domain or constrained task, like problem-solving Naeini et al. (2023); Tian et al. (2024), humor Zhong et al. (2024), or simile generation He et al. (2023), where creativity is just one of several assessed aspects. Unlike open-domain tasks, they are often entangled with other concepts, making it hard to isolate and generalize creativity itself to other domains, such as literature. Second (**granularity**), most methods evaluate creativity at the model or subject level rather than at the level of individual responses Mednick & Halpern (1968); Torrance (1966). While useful for comparing models, they struggle to distinguish which of two responses to the same prompt is more creative Chakrabarty et al. (2024); Zhao et al. (2025). We refer to the latter as text-level creativity (or simply *text creativity*). It is especially valuable as it highlights specific responses for improvement, providing more actionable insights than coarse model- or subject-level evaluations.

---

✉ Corresponding author.

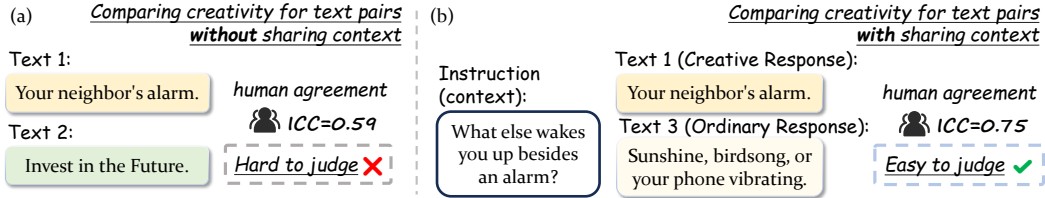

Figure 1: An example of how to formulate the problem of text creativity evaluation to better evaluate.

Third, (**effective automation**) automating cross-domain creativity evaluation reduces human effort and supports iterative improvement. LLMs have shown effectiveness as automatic evaluators, in areas such as helpfulness and coherence Hu et al. (2024b); Kim et al. (2024); Li et al. (2024), known as LLM-as-a-judge Gu et al. (2024); Li et al. (2025a); Zheng et al. (2023). However, creativity evaluation remains underexplored. While early attempts prompt LLMs to assess creativity Summers-Stay et al. (2023); Zhao et al. (2025), leveraging the cross-domain strengths of advanced models like GPT-4o Hurst et al. (2024), their judgments often suffer from unreliability Chakrabarty et al. (2024), inconsistency Wang et al. (2024a), and high cost Chen et al. (2023).

This paper aims to address these issues by proposing a novel evaluation methodology for automated, cross-domain creativity assessment, including a cross-domain benchmark dataset labeled by 30 human judges and an effective LLM-based creativity evaluator. Developing this framework presents two key challenges. The first is how to facilitate consistent human labeling. We observe that without clear contextual guidance, human annotators may struggle to reach consistent judgments, since creativity may be understood differently in different contexts. For example, as shown in Figure 1 (a), when three annotators independently rated 400 decontextualized text pairs, the agreement among them was only moderate (with an Intraclass Correlation Coefficient, *i.e.*, ICC, of 0.59). The second challenge is how to train a reliable LLM evaluator given the scarcity of creative data. Data scarcity limits the ability of evaluators to generalize across diverse domains and their effectiveness. To address this, it is thus crucial to collect large-scale training data in a weakly supervised manner.

Our work resolves these two challenges by introducing a framework that generates multiple creative responses conditioned on the same context. On the one hand, this setup ensures high-quality human annotations of text creativity pairs. As shown in Figure 1 (b), when a shared instruction was provided as a context, the agreement improved significantly (ICC increases to a good level of 0.75). On the other hand, by controlling the response generation process for the same context, we automatically generate large-scale pseudo labels for their creativity levels in a weakly supervised manner, which solves the data scarcity issue. Specifically, our contributions are as follows:

• We propose a **context-aware, pairwise comparison-based** evaluation protocol for assessing text creativity. Using this protocol, we manually annotate a test set of **over 3,000** samples to benchmark text creativity evaluators. Notably, even state-of-the-art LLMs perform poorly on it compared to humans, underscoring a key performance bottleneck in current evaluators. To support training, we further construct **CreataSet**, a large-scale dataset incorporating creative tasks in **87 domains**, including over **1M instruction-response pairs** with varying weakly supervised creativity levels.

• Building on CreataSet, we introduce **CrEval**, an LLM-based creativity evaluator. To the best of our knowledge, our work is among the first to evaluate creativity across multiple domains using pairwise assessments. CrEval outperforms strong frontier models, *e.g.*, GPT-4o by **18.7%** in agreement with human judges, and demonstrates strong domain generalization capabilities. We further show that CrEval can enhance LLM creativity, offering a practical approach to improve generative AI.

## 2    RELATED WORK

**Creativity Evaluation**    Evaluating creativity has been a long-standing challenge Kim (2006); Acar & Runco (2019). Many proposed methods Gray et al. (2019); Zhao et al. (2025); Beketayev & Runco (2016); Sun et al. (2025) adopt frameworks targeting particular tasks, such as the Remote Associates Test (RAT) Mednick & Halpern (1968) or the Torrance Test of Creative Thinking (TTCT) Torrance (1966), which measures human divergent thinking through scoring ideas on fluency, originality,

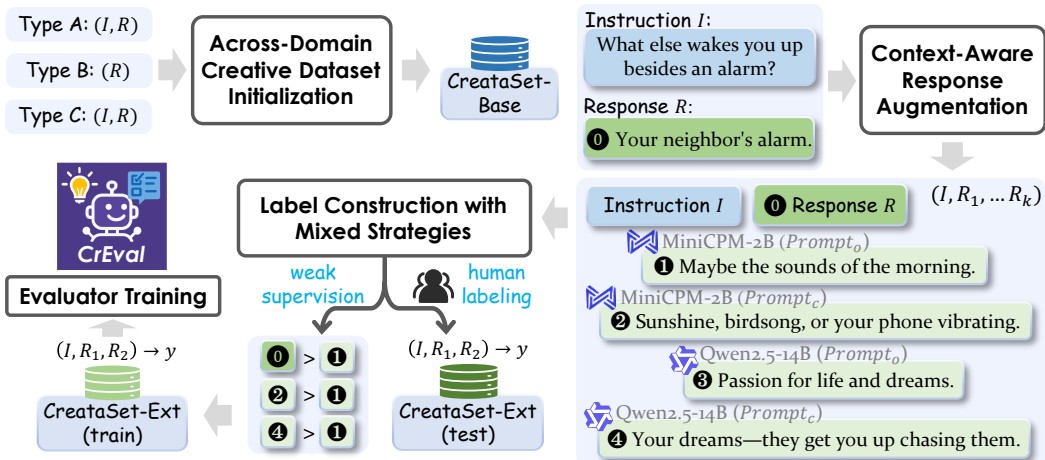

Figure 2: The construction process of CreataSet and training process of CrEval.

flexibility, and elaboration (*e.g.*, listing diverse uses for a paperclip). Subsequent adaptations have applied TTCT principles to creative writing (TTCW) Chakrabarty et al. (2024); Li et al. (2025c) or to evaluate LLMs on such tasks Zhao et al. (2025), while other work uses problem-solving as a creativity proxy Naeini et al. (2023); Tian et al. (2024). However, these approaches are often narrow in scope, focusing on a limited set of tasks and primarily assessing a model's creative ability rather than the creativity of the generated text itself.

Existing methods for evaluating textual creativity face significant limitations. Heuristic scoring He et al. (2023), matching for unique n-grams on a reference corpus (Creativity Index) Lu et al. (2025b), and calculating divergent semantic integration using BERT Devlin et al. (2019) (DSI) Johnson et al. (2023) are often constrained by their specific designs, reliance on static corpora, and limited generalizability. While prompting general-purpose LLMs (*e.g.*, GPT-4) has become common Summers-Stay et al. (2023); Zhao et al. (2025); Chakrabarty et al. (2024; 2025), results are often unsatisfactory Olson et al. (2024); Chakrabarty et al. (2024; 2025); Lu et al. (2025a). Another work, LitBench Fein et al. (2025), has trained reward models on specialized preference data Fein et al. (2025), but their applicability remains confined to creative writing, lacking broader generalization to other domains. Consequently, reliable evaluation of textual creativity still depends heavily on costly and inefficient human judgment, such as the Consensual Assessment Technique (CAT) Baer & McKool (2009); Marco et al. (2024), which cannot provide automated feedback for improving models. To overcome these challenges, we propose a novel approach that leverages the LLM-as-a-judge paradigm for more efficient and accurate creativity assessment.

**LLM-as-a-Judge** In the area of automatic evaluation for text generation, recent advent of large language models (LLMs) has enabled the evaluation paradigms to incorporate LLMs to be more accurate and flexible Gao et al. (2025), known as LLM-as-a-judge Zheng et al. (2023); Gu et al. (2024); Li et al. (2025a), capable of assessing more diverse dimensions of text quality. Prior works focus more on the evaluation of text attributes like relevance Liu et al. (2023); Abbasiantaeb et al. (2024); Liu et al. (2024), helpfulness Kim et al. (2024); Li et al. (2024), or overall excellence Dongfu et al. (2024); Hu et al. (2024b), etc. Other works also explore how to adapt LLMs to evaluate specific domains such as code generation Tong & Zhang (2024); Wu et al. (2025a) and dialogue generation Lin & Chen (2023); Zhang et al. (2024). However, few of these works investigate how to evaluate text creativity, making it hard to assess and improve the creative aspects of text generation. In our work, we propose to assess it in a pairwise-comparison manner, and provide a comprehensive study on leveraging LLMs to evaluate text creativity.

## 3 METHODOLOGY

In this section, we employ a three-step process to construct our large-scale weakly supervised dataset CreataSet to support our evaluation protocol and train CrEval. First, in **Across-Domain Creativity**

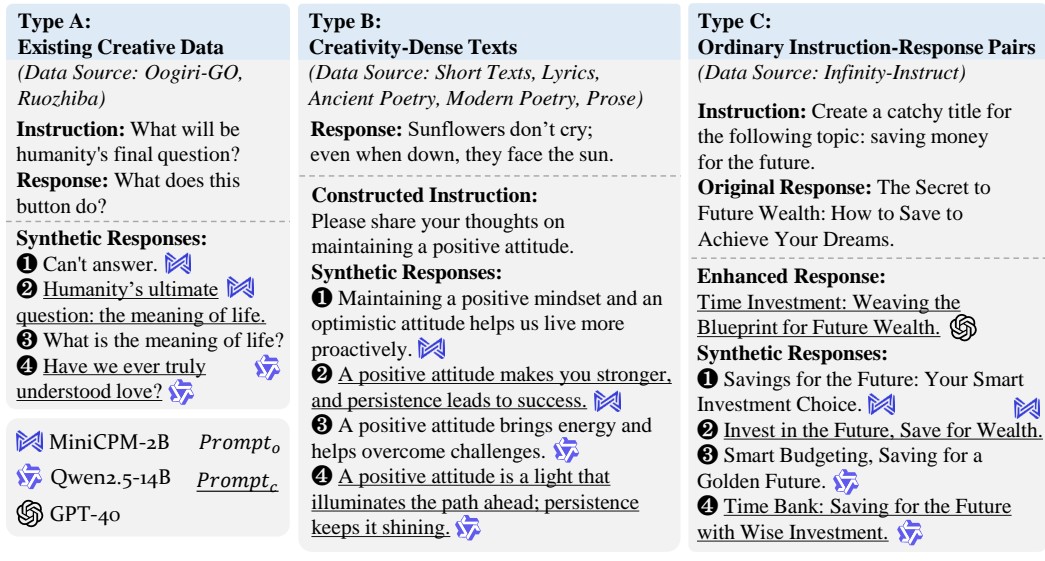

Figure 3: The examples of three different types of data. The original data are above the dashed line, while our constructed components are below.

**Dataset Initialization**, we gather initial data in 87 diverse domains with varying lengths, generating corresponding instructions to create initial instruction-response pairs $(I, R)$. Second, **Context-Aware Response Augmentation** expands these pairs by generating responses of varying creative levels for the same instruction $I$. Finally, in **Label Construction with Mixed Strategy**, we pair the responses and assign a label $y$, yielding training samples of the form $(I, R_1, R_2, y)$ for the creativity evaluator CrEval. For meta-evaluation, we manually annotate a test set to benchmark CrEval against other evaluators. The overall data construction pipeline is illustrated in Figure 2.

### 3.1 ACROSS-DOMAIN CREATIVITY DATASET INITIALIZATION

To build a creativity dataset across diverse domains, we gather initial data with varying creativity levels from eight sources. We unified them into a consistent $(I, R)$ format.

**Multi-Domain Multi-Source Data Collection** We aim to collect data from diverse sources and domains to construct a broad distribution in both domain coverage and response length, thereby enabling the model to generalize across a wide range of scenarios. Specifically, we begin by collecting data from existing creativity datasets, such as Oogiri-GO Zhong et al. (2024) and Ruozhiba Bai et al. (2025), which naturally contain creative $(I, R)$ pairs in the humor domain (Type A in Figure 3). We further incorporate creativity-dense texts $(R)$ from corpora of human creative works, such as poetry, lyrics, and prose, sourced from well-known websites[1]. To enhance length diversity, we curate a sub-dataset called *Short Texts*, comprising inspiring and thought-provoking sentences collected from online sources[2]. Most of these entries consist of standalone texts $(R)$ without explicit input prompts (Type B in Figure 3). In addition, aiming to capture data with diverse creativity levels and expand domain coverage, we leverage an existing instruction-tuning dataset Infinity-Instruct Li et al. (2025b), given its high-quality $(I, R)$ pairs spanning a wide range of domains (Type C in Figure 3).

**Unified Instruction-Response Standardization** To standardize the multi-source data into a unified $(I, R)$ format, we first enrich standalone texts by generating missing instructions. We train an instruction generator by reversing an instruction-tuning dataset (Infinity-Instruct). The generator learns to produce an instruction $I$ given a response $R$. We generate an instruction for each standalone text, thus forming $(I, R)$ pairs. To prevent non-creative data from obscuring creative data, we followed previous work Ritchie (2007) and applied some filters. All data are ultimately

---

[1]https://github.com/chinese-poetry/chinese-poetry, https://github.com/VMIJUNV/chinese-poetry-and-prose, https://github.com/yuxqiu/modern-poetry, https://music.163.com, https://m.sbkk8.com

[2]https://www.juzikong.com/

| Dataset | Cross-domain | Granularity | Auto-Evaluator | Total Words | # Samples | Train/Test |
|---|---|---|---|---|---|---|
| Oorigi-GO Zhong et al. (2024) | ✗ (humor) | Subject Level | ✗ | 894,712 | 15,797 | train & test |
| MacGyver Tian et al. (2024) | ✗ (problem-solving) | Subject Level | ✗ | 249,385 | 1,683 | test |
| DPT Jr. et al. (2025) | ✗ (problem-solving) | Subject Level | ✗ | 12,576 | 803 | test |
| TTCW Chakrabarty et al. (2024) | ✗ (creative writing) | Subject Level | ✗ | 58,426 | 48 | test |
| Creative Writing v3 Paech (2023) | ✗ (creative writing) | Subject Level | ✗ | 10,176 | 32 | test |
| TTCT+ Zhao et al. (2025) | ✓ (7 domains) | Subject Level | ✗ | - | 700 | test |
| LitBench Fein et al. (2025) | ✗ (creative writing) | Individual Text Level | ✓ | 16,309,661 | 43,827 | train & test |
| WritingBench Wu et al. (2025b) | ✓ (100 domains) | Individual Text Level | ✓ | 1,875,146 | 1,000 | test |
| CreataSet-Base (ours) | ✓ (87 domains) | Individual Text Level | ✓ | 20,720,179 | 112,965 | train & test |

Table 1: The statistics of different creative datasets. Auto-Evaluator denotes whether an automatic evaluator is proposed based on this dataset. TTCT+ and training data for the evaluator in Writing-Bench are not publicly available. We calculate the total word count of the responses of each dataset.

formatted as $(I, R)$ pairs (as shown in Figure 3, forming **CreataSet-Base**, with over 113k creative samples. Due to the deeply contextual nature of creativity, which is highly subject to cultural and linguistic context, the dataset is predominantly in Simplified Chinese. However, our framework is language-agnostic and can be easily extended to other languages.

Table 1 compares CreataSet-Base with other creativity-related datasets, highlighting its larger scale. To assess domain diversity (*i.e.*, thematic category), we followed prior works Tian et al. (2024); Wang et al. (2023); Jin et al. (2024) and started from a manually curated seed taxonomy. We then adopted GPT-4o-mini to classify each data sample into a fine-grained category, yielding 87 distinct subdomains. They were then aggregated by the model into broader, semantically coherent ones, resulting in 17 core domains. The distribution of these domains is shown in Figure 4. Additional details on response length and semantic distributions are provided in Appendix A.3.

Figure 4: Domain distribution of CreataSet-Base. Secondary domains for the top 5 primary ones are shown in gray.

### 3.2 CONTEXT-AWARE RESPONSE AUGMENTATION

Before constructing pairwise data $(I, R_1, R_2)$ for training the evaluator, we first augment the set of responses for each instruction, *i.e.*, $(I, R_1, \ldots, R_k)$. This aims to enrich the creative diversity, enabling the construction of pairs with creative differences. To efficiently construct such data at scale, we employ open-sourced models with different levels of capability, *e.g.*, Qwen2.5-14B-Instruct Yang et al. (2024) and MiniCPM-2B-SFT Hu et al. (2024a), to generate responses for instructions in CreataSet-Base, as illustrated in Figure 2. For each model, we use two prompting modes to induce varying creativity levels: (1) $\texttt{Prompt}_o$, a general prompt that elicits ordinary responses; and (2) $\texttt{Prompt}_c$, a creativity-oriented prompt that encourages more imaginative outputs. By adopting different models/prompts, we generate multiple synthetic responses. The original responses in Type C data are direct answers to instructions with weak creativity. To enrich those, we further prompt GPT-4o to generate more creative ones to the same instructions. Finally, we name this dataset in the form $(I, R_1, \ldots, R_k)$ as **CreataSet-Ext**. The diversity analysis of augmented responses and the prompts used are in Appendix A.3.5 and Appendix A.6, respectively.

### 3.3 LABEL CONSTRUCTION WITH MIXED STRATEGIES

We combine responses into pairs $(I, R_1, R_2)$ and use a mixed strategy to assign labels for training and testing separately, since the label requirements differ between them. Reliable human-annotated labels are essential for meta-evaluation to accurately assess model performance, while constructing labels in a large scale is more important for training. We detail them in the following.

**High-Quality Human Labeling for Test Benchmark Construction** To ensure diversity in the test set, we sample 50 instances from each data source in CreataSet, yielding 400 initial samples. These are further augmented using GPT-4o-mini with both prompts to enhance the distribution dif-

ference for evaluation. Following prior work Weinstein et al. (2022); Johnson et al. (2023), we recruited 30 qualified annotators from 18 different majors to rate response creativity on a 4-point Likert scale, with responses presented in randomized order (more details are in Appendix A.8). Each response's creativity score is computed as the average of all ratings. The annotations exhibit high inter-rater reliability (Intraclass Correlation Coefficient, ICC(2k)=0.92). Finally, we construct a 3K test set in the format $(I, R_1, R_2, y)$, where pairs with score differences $> 0.3$ are labeled as *distinguishable*, and those with differences $< 0.1$ as *comparable* (tie).

**Weakly-Supervised Pseudo Labels for Training Set Construction**    For the training set, we assign weakly supervised pseudo-labels to response pairs in CreataSet-Base, enabling large-scale label construction. Our approach is based on two key assumptions: (1) *stronger models tend to produce more creative responses than weaker ones*, and (2) *creativity-focused prompts elicit more creative outputs than ordinary prompts*.

To validate these assumptions, we sampled 150 data groups $((I, R_1, \ldots, R_k))$ with 1,050 response pairs for each model/prompt combination and recruited 3 annotators to compare their creativity. The results show that creativity distinctions based on assumption (1) achieve 86.6% accuracy, and assumption (2) achieves 81.4%, confirming the reliability of both heuristics. For creatively comparable samples (the tie cases), we randomly pair responses produced by the same models using $\texttt{Prompt}_o$. Using these assumptions, we assign labels $y$ to response pairs in CreataSet-Ext, resulting in training data of the form $(I, R_1, R_2, y)$.

## 3.4    CrEval Training

The constructed large-scale CreataSet-Ext can enable us to train CrEval. It provides triplets $(I, R_1, R_2) \in \mathcal{D}$ as input, and trained to minimize the classification loss:

$$\mathcal{L} = - \sum_{(I, R_1, R_2) \in \mathcal{D}} \log P(y|I, R_1, R_2), \tag{1}$$

where $P(y|I, R_1, R_2)$ represents the probability of the label $y$ given the triplet $(I, R_1, R_2)^3$. To mitigate the positional bias, we follow previous works Wang et al. (2024a;b); Li et al. (2024) by augmenting the data by swapping $R_1$ and $R_2$ in the input and adjusting the corresponding label. Additionally, we apply negative sampling by randomly selecting a response to serve as the least creative response, further enhancing the model's awareness of the instruction context $I$.

During inference, the model predicts whether $R_1$ is more creative than $R_2$, vice versa, or if they are creatively comparable. Moreover, a reference response $R^r$, generated by either a human or a model, can be a baseline for comparing the creativity of another response $R$ in such a comparison manner.

## 4    Experiments

### 4.1    Experimental Setup

In our experiments, we set $k = 5$ in response augmentation. This is shared across all data sources. Based on our human-labeled test set of CreataSet, we adopt *F1 score*, *Kappa score*, and *Agreement rate* to evaluate the performance of different methods, following previous work Wang et al. (2024b); Li et al. (2024). All metrics are calculated twice by swapping the order of the two responses, and then the average scores are reported. Following Hu et al. (2024b), to eliminate the influence of sampling randomness, we set the temperature T to 0 for deterministic results, while other methods retain their original settings. We conduct pairwise comparison experiments on CreataSet where CrEval is compared with the following baselines:

**Traditional Metrics:** (1) **Perplexity (PPL)**: A simple baseline where we use Qwen2.5-7B-Instruct to calculate the perplexity of a response. Higher perplexity indicates higher novelty and creativity. (2) **Divergent semantic integration (DSI)** Johnson et al. (2023): It adopts BERT Devlin et al. (2019) to calculate the average semantic distance between all words in the response. A higher DSI indicates higher creativity. (3) **Creativity Index** Lu et al. (2025b): A corpus-based metric calculates creativity inversely to n-gram similarity with a reference corpus.

---

[3]Since we use LLM backbones, the classification label is treated as a text output conditioned on the prompt.

| Method | S.T. | Lyr. | A.P. | M.P. | Pro. | Oog. | Ruo. | Inf. | Average F1 | Kappa | Agree. |
|---|---|---|---|---|---|---|---|---|---|---|---|
| *Traditional Metrics* | | | | | | | | | | | |
| PPL | 0.464 | 0.245 | 0.245 | 0.316 | 0.349 | 0.515 | 0.329 | 0.374 | 0.357 | -0.042 | 0.430 |
| DSI | 0.440 | **0.430** | **0.354** | **0.527** | 0.377 | 0.578 | 0.561 | 0.528 | 0.480 | 0.175 | 0.457 |
| Creativity Index | **0.695** | 0.368 | 0.338 | 0.417 | **0.592** | **0.585** | **0.566** | **0.640** | **0.531** | **0.231** | **0.568** |
| *Frontier LLMs* | | | | | | | | | | | |
| o3 | 0.802 | 0.589 | 0.596 | 0.667 | 0.663 | **0.774** | 0.832 | 0.769 | 0.721 | 0.578 | 0.725 |
| o1 | **0.807** | 0.573 | 0.629 | 0.670 | 0.672 | 0.738 | 0.790 | 0.798 | 0.720 | 0.563 | 0.664 |
| GPT-4o | 0.800 | **0.605** | 0.641 | **0.699** | 0.667 | 0.749 | 0.633 | 0.789 | 0.703 | 0.519 | 0.642 |
| GPT-3.5 | 0.686 | 0.486 | 0.425 | 0.548 | 0.489 | 0.667 | 0.567 | 0.743 | 0.585 | 0.350 | 0.522 |
| DeepSeek-R1 | 0.743 | 0.479 | 0.494 | 0.578 | 0.612 | 0.751 | 0.745 | 0.733 | 0.653 | 0.457 | 0.547 |
| DeepSeek-V3 | 0.780 | 0.584 | 0.584 | 0.681 | 0.684 | 0.765 | 0.774 | 0.784 | 0.714 | 0.558 | 0.668 |
| Claude-3.5-Sonnet | 0.775 | 0.603 | 0.634 | 0.671 | **0.702** | 0.762 | 0.850 | **0.810** | 0.727 | **0.609** | **0.740** |
| Claude-3.5-Haiku | 0.748 | 0.573 | 0.509 | 0.633 | 0.652 | 0.724 | 0.695 | 0.779 | 0.669 | 0.496 | 0.641 |
| Gemini-2.5-Pro | 0.764 | 0.569 | 0.585 | 0.639 | 0.656 | 0.760 | **0.866** | 0.752 | 0.708 | 0.557 | 0.702 |
| Gemini-2.5-Flash | 0.785 | 0.588 | **0.642** | 0.670 | 0.692 | 0.761 | 0.858 | 0.797 | **0.731** | 0.582 | 0.682 |
| G-Eval (GPT-4o) | 0.772 | 0.583 | 0.568 | 0.665 | 0.694 | 0.759 | 0.803 | 0.793 | 0.712 | 0.558 | 0.677 |
| G-Eval (GPT-3.5) | 0.636 | 0.494 | 0.460 | 0.561 | 0.493 | 0.608 | 0.575 | 0.774 | 0.582 | 0.339 | 0.500 |
| *7B Scale LLMs* | | | | | | | | | | | |
| Gemma-2-9B-it | **0.795** | **0.562** | 0.619 | 0.654 | 0.646 | 0.751 | 0.779 | 0.788 | 0.704 | 0.544 | 0.654 |
| LLaMA3.1-8B-Instruct | 0.713 | 0.548 | 0.440 | 0.618 | 0.615 | 0.649 | 0.573 | 0.782 | 0.621 | 0.418 | 0.565 |
| PandaLM-7B | 0.390 | 0.435 | 0.454 | 0.469 | 0.346 | 0.398 | 0.540 | 0.506 | 0.453 | 0.129 | 0.326 |
| Prometheus-7B | 0.330 | 0.365 | 0.326 | 0.315 | 0.342 | 0.369 | 0.449 | 0.498 | 0.377 | 0.097 | 0.352 |
| AUTO-J | 0.659 | 0.526 | 0.377 | 0.561 | 0.541 | 0.553 | 0.565 | 0.720 | 0.567 | 0.323 | 0.512 |
| WritingBench-Critic | 0.715 | 0.528 | 0.500 | 0.626 | 0.548 | 0.600 | 0.641 | 0.712 | 0.612 | 0.362 | 0.576 |
| Qwen2.5-7B-Instruct | 0.710 | 0.494 | 0.426 | 0.578 | 0.487 | 0.647 | 0.704 | 0.771 | 0.614 | 0.403 | 0.574 |
| **CrEval-7B (ours)** | 0.779 | 0.556 | **0.649** | **0.681** | **0.665** | **0.778** | **0.873** | **0.820** | **0.732** | **0.601** | **0.745** |
| Δ (v.s. base model) | +9.7% | +12.6% | +52.3% | +17.8% | +36.6% | +20.2% | +24.0% | +6.4% | +19.2% | +49.1% | +29.8% |
| *13B Scale and Larger LLMs* | | | | | | | | | | | |
| Qwen2.5-72B-Instruct | 0.751 | 0.558 | 0.520 | 0.655 | 0.594 | 0.734 | 0.833 | 0.806 | 0.692 | 0.535 | 0.673 |
| LLaMA3.1-70B-Instruct | 0.736 | 0.564 | 0.559 | 0.642 | 0.624 | 0.732 | 0.764 | **0.810** | 0.684 | 0.535 | 0.675 |
| Gemma-3-27B-it | **0.792** | **0.572** | 0.608 | 0.650 | 0.666 | 0.753 | 0.789 | 0.783 | 0.706 | 0.564 | 0.702 |
| Gemma-3-12B-it | 0.761 | 0.542 | 0.575 | 0.615 | **0.674** | 0.729 | 0.667 | 0.772 | 0.672 | 0.498 | 0.633 |
| Prometheus-13B | 0.445 | 0.377 | 0.329 | 0.372 | 0.410 | 0.386 | 0.367 | 0.641 | 0.416 | 0.095 | 0.400 |
| Qwen2.5-14B-Instruct | 0.742 | 0.568 | 0.523 | 0.629 | 0.629 | 0.717 | 0.783 | 0.797 | 0.683 | 0.523 | 0.661 |
| **CrEval-14B (ours)** | 0.786 | 0.556 | **0.650** | **0.680** | 0.672 | **0.797** | **0.882** | **0.810** | **0.735** | **0.613** | **0.762** |
| Δ (v.s. base model) | +5.9% | −2.1% | +24.3% | +4.8% | +6.8% | +11.2% | +12.6% | +1.6% | +7.6% | +17.2% | +15.3% |

Table 2: Results of different methods on our CreataSet test set. Best results in the same group are highlighted in bold, and the second-best are underlined. S.T., Lyr., A.P., M.P., Pro., Oog., Ruo., and Inf. represent Short Texts, Lyrics, Ancient Poetry, Modern Poetry, Prose, Oogiri-Go, Ruozhiba, and Infinity-Instruct, respectively. We gray out the results of frontier LLMs due to their larger sizes.

**Evaluation-Centric Models:** Several evaluation-centric models including prompting-based **G-Eval** Liu et al. (2023) and fine-tuned LLMs **PandaLM** Wang et al. (2024b), **Prometheus** Kim et al. (2024), **AUTO-J** Li et al. (2024) and **WritingBench-Critic** Wu et al. (2025b).

**General-purpose LLMs as Evaluators:** We compare CrEval against several general-purpose LLMs, including **LLaMA3.1-{8,70}B-Instruct** Dubey et al. (2024), **Gemma-{2-9B,3-12B,3-27B}-it** Rivière et al. (2024); Kamath et al. (2025), **Qwen2.5-{7,14,72}B-Instruct** Yang et al. (2024), **GPT-3.5-Turbo-1106** OpenAI (2022), **GPT-4o** Hurst et al. (2024), **OpenAI o1/o3** Jaech et al. (2024); OpenAI (2025), **DeepSeek-{V3,R1}** DeepSeek-AI et al. (2024; 2025), **Claude-3.5-{Haiku, Sonnet}** Anthropic (2024a;b), and **Gemini-2.5-{Flash, Pro}** Comanici et al. (2025). We evaluate all models using the same prompt as for CrEval.

## 4.2 HOW WELL CAN CrEval SIMULATE HUMAN EVALUATION?

As shown in Table 2, CrEval demonstrates consistent and significant improvements over all baselines across the evaluated metrics. Notably, the 14B variant even surpasses most frontier baselines, improving F1 by 2.9%, Kappa by 9.7%, and agreement rate by 12.6% compared to strong competitors like DeepSeek-V3. These results validate the effectiveness of our approach for simulating human creativity assessment. Second, traditional metrics such as PPL and DSI perform poorly; *e.g.*, PPL yields a Kappa score near zero, indicating their weak correlation with human judgment. The Creativity Index metric improves on them but remains limited by its reliance on n-gram matching and fails to capture the semantic creativity conveyed through conventional lexical choices. Third, while Gemma models achieve the highest F1 scores of the according groups on Short Texts and

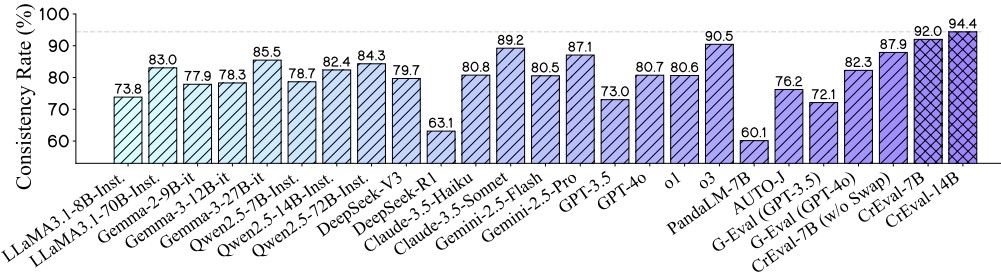

Figure 5: Consistency rate of different methods when swapping the order of responses. We inlcude an ablation version CrEval-7B (w/o Swap) without explicitly swapping response positions.

| Method | F1 | Kappa | Agreement |
|---|---|---|---|
| CrEval-7B | 0.732 | 0.601 | 0.745 |
| w/o Neg. | 0.723 | 0.586 | 0.745 |
| w/o Syn. | 0.665 | 0.464 | 0.634 |
| w/ Only Syn. | 0.585 | 0.356 | 0.589 |

Figure 6: Evaluation results of ablation study on different data components.

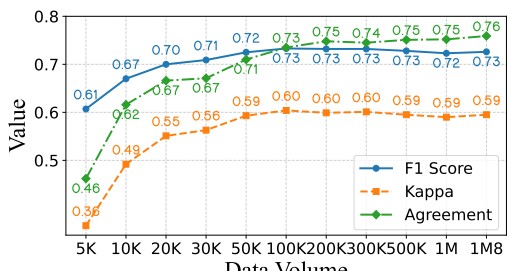

Figure 7: Performance variation with data scales.

Lyrics, they struggle to generalize across other creative domains like humor (*e.g.*, Oogiri-Go and Ruozhiba) and ancient genres. Claude-3.5-Sonnet excels in evaluating Prose, indicating a stronger capacity for assessing creativity in longer texts. In contrast, CrEval exhibits more balanced and robust performance across all creative domains.

LLMs may favor certain positions of the response, known as positional bias Wang et al. (2024a), which may lead to inconsistent evaluation results when swapping the order of responses. We have conducted a consistency analysis to evaluate the stability of different methods, inlcuding comparing with an ablation version CrEval (w/o Swap) where CrEval was trained without explicitly balancing the positions. As shown in 5, CrEval achieves the highest consistency rate of 94.4, indicating that it is more consistent and reliable in evaluating creativity compared to other methods. Also, omitting position swapping during training introduces a position bias and leads to decreased performance.

### 4.3 How Do Data Influence CrEval?

**Data Composition.** In training CrEval, we use $(I, R_1, R_2, y)$ of different pseudo-creativity levels. To investigate their influence, we conduct an ablation study by training multiple CrEval variants with different data compositions as follows: (1) **CrEval-w/o Neg.**: Training CrEval without sampling negative responses. (2) **CrEval-w/o Syn.**: Training CrEval with only the original responses (highest creativity) in CreataSet without synthetic ones. (3) **CrEval-w/ Only Syn.**: Training CrEval with only synthetic responses (lower creativity) in CreataSet without original ones. Table 6 presents the ablation study results. The results indicate that each type of data makes a positive contribution. The original human-created responses contribute the most, as they provide diverse, high-quality information that better aligns CrEval with human preferences. Synthetic data plays a crucial role in helping the model grasp the characteristics of creative responses, particularly those that LLMs can generate. Meanwhile, negative responses offer additional information to improve the model's ability to measure the relevance between responses and instructions.

**Data Scale.** To assess the impact of data volume, we train CrEval on datasets of varying scales, shown in Figure 7. F1, Kappa, and agreement rates improve with data size but plateau after 100K samples. This suggests that while more data benefits CrEval, the gains diminish at higher scales.

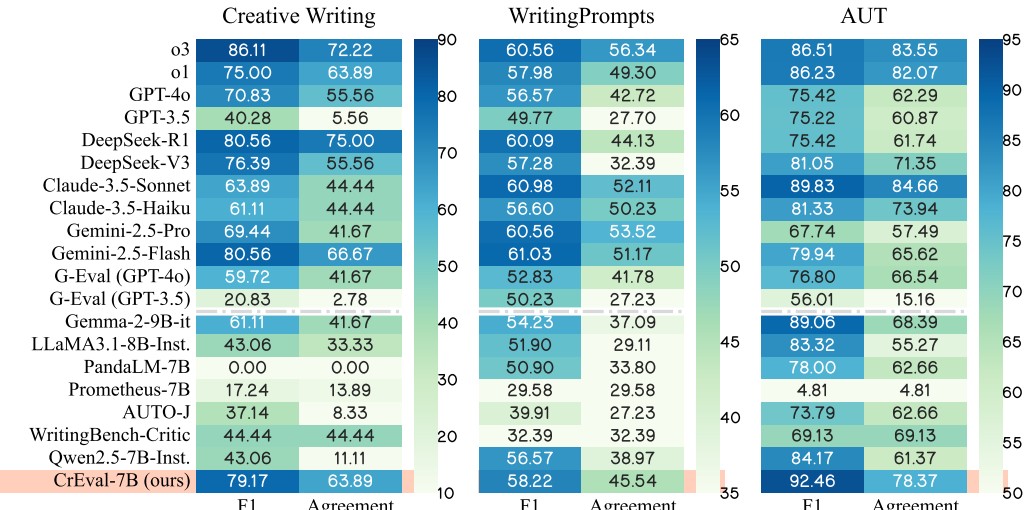

Figure 8: Agreement and F1 scores on three O.O.D. datasets.

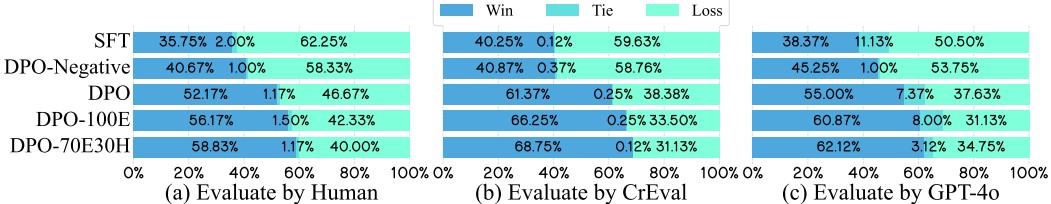

Figure 9: Win rate of different methods over GPT-4o-mini responses. DPO-Negative denotes DPO with negative sampled responses as reject samples. DPO-100E and DPO-70E30H use all easy and 70% easy+30% hard responses as reject samples, respectively.

## 4.4 DOES CRE VAL DEMONSTRATE OUT-OF-DISTRIBUTION GENERALIZATION?

Due to the scarcity of human-annotated creative pairs, finding suitable out-of-distribution (O.O.D.) datasets for meta-evaluation remains challenging. To address this, we conduct three O.O.D. experiments on two *creative writing* datasets and one classical creativity benchmark, the *Alternative Uses Task (AUT)*. For creative writing, we adopt data from Chakrabarty et al. (2024), which contains long responses produced by both humans and models. Following their findings, we treat human responses as more creative and construct 36 evaluation pairs. Besides, we curate 213 pairs from another *WritingPrompts* dataset, selecting samples with a like-count difference greater than 10 as a proxy for creativity. For the AUT task, we use the dataset from Sun et al. (2023), focusing on the annotated alternative uses for "bowl". We form pairs from responses whose human-assigned creativity scores differ by more than two points, resulting in 541 test pairs.

As shown in Figure 8, CrEval achieves the best performance among models of similar scale (~7B) and even outperforms much larger frontier models like GPT-4o and DeepSeek-V3. This strong generalization ability can be attributed to its robust training on diverse and creative text sources, enabling it better to capture subtle qualitative differences in open-ended generation tasks. The consistent advantage across both datasets underscores its effectiveness in text creativity evaluation.

## 4.5 CAN CRE VAL ENHANCE MODEL CREATIVITY?

As a creativity evaluator, CrEval can differentiate response creativity, allowing us to leverage it for enhancing model creativity. We randomly sample 10K data from CreataSet to train Qwen2.5-7B-Instruct, using the original response as the ground truth, serving as the standard (the **SFT** baseline). By utilizing synthetic candidate responses (randomly sampled), we apply DPO Rafailov et al. (2023)

to take low-creativity responses as reject samples (the **DPO** baseline). We also randomly sample negative responses from other instructions as reject samples, denoted as **DPO-Negative**.

Given an instruction and multiple candidate responses, CrEval performs pairwise creativity comparisons, scoring wins as 3 points, ties as 1, and losses as 0. This scoring yields a creativity ranking, with the top-ranked responses as *hard* and the lowest as *easy* samples. We control creativity difficulty by adjusting the hard/easy ratio in DPO rejections, evaluating methods on the CreataSet test set, with win rates against GPT-4o-mini measured by CrEval, GPT-4o, and human annotators.

Results in Figure 9 demonstrate that DPO yields significant gains over SFT in all evaluation settings (CrEval, GPT-4o, human). The inferior performance of DPO-Negative relative to DPO demonstrates that contextual conditioning is crucial for accurate creativity assessment. Leveraging CrEval for creativity-aware data selection leads to further improvements, with DPO-70E30H achieving the highest win rate. DPO-100E, which treats all easy samples as rejections, shows marginal improvement, indicating that a clearer distinction between chosen and rejected examples is crucial for learning creativity. DPO-70E30H achieves the highest win rate by using 30% hard samples as rejections, underscoring the benefit of a balanced mixture of creativity difficulty levels.

## 5 CONCLUSION

In this paper, we propose a novel pairwise-comparison framework for evaluating textual creativity and present CreataSet, a large-scale dataset across diverse domains. Based on it, we develop CrEval, an LLM-based evaluator that significantly outperforms existing methods in alignment with human judgments. Our experiments highlight the essential role of combining both human and synthetic data in training robust creativity evaluators, and demonstrate that CrEval exhibits out-of-distribution generalization. We further find the practical value of integrating CrEval into generation pipelines to boost LLM creativity. We believe that CreataSet and CrEval will be valuable assets for the research community, driving progress toward more accurate and scalable creativity evaluation.

### ETHICS STATEMENT

This work adheres to the ICLR Code of Ethics. The study involved human participants who were recruited as data annotators, and the annotation tasks were designed to be of minimal risk. The study did not involve any animal experimentation. All datasets used were sourced in compliance with relevant usage guidelines, with appropriate measures taken to protect privacy and prevent any unauthorized use of data. We have strived to mitigate potential biases and avoid discriminatory outcomes throughout the research process. No personally identifiable information was utilized, and no experiments were conducted that would raise ethical, privacy, or security concerns. We are committed to upholding principles of transparency and integrity in all aspects of this research.

### REPRODUCIBILITY STATEMENT

We have taken comprehensive steps to ensure the reproducibility of the results presented in this paper. We have made our code, datasets and models publicly available. The experimental setup, including training procedures and model configurations, is described in detail. We believe these steps will enable the community to validate and build upon our work effectively.

### ACKNOWLEDGMENTS

This work is supported by the National Key R&D Program of China (2023YFF0905402) and Kuaishou Technology. We acknowledge the reviewers for their insightful comments.

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

# A  APPENDIX

## A.1  SUBJECTIVITY AND IMPORTANCE OF CREATIVITY

While creativity is inherently subjective, we wish to highlight that:

(1) **Creativity is a core AI objective, both historically and practically, whose complexity should be embraced rather than avoided.** "Randomness and creativity" was identified as one of the seven key problems at the Dartmouth Summer Research Project on Artificial Intelligence and recognized as essential to human-level intelligence and central to the very definition of machine intelligence McCarthy et al. (2006). With the rise of LLMs in domains such as storytelling, ideation, and design, evaluating creative ability has become both timely and necessary. Without creativity, AI models cannot generate truly novel, out-of-domain ideas — a crucial aspect of human-level intelligence that has shaped modern civilization.

(2) **The subjectivity of creativity is inevitable but controllable through our rigorous evaluation design.** Although creativity involves personal and cultural variation, psychological studies Barbot et al. (2019); Parkhurst (1999) show that people often converge in recognizing creative content. For instance, more creative ideas, such as the novel *Harry Potter*, typically receive higher engagement (*e.g.*, more likes). Rather than eliminating subjectivity, our goal is to model shared human judgment in a reproducible manner using pairwise comparisons and consensus-driven aggregation. To this end, we collaborate with annotators from diverse backgrounds to ensure that the resulting evaluation framework captures a robust, collective understanding of creativity.

(3) **Our goal is not to equate creativity with conformity, but rather to approximate shared human judgments in a reproducible and scalable way.** Psychological studies have shown humans can reliably recognize creativity across cultures and domains, especially when it combines novelty with usefulness Runco & Jaeger (2012); Barbot et al. (2019). Our use of consensus aims to reflect this shared intuition, not to suppress unconventionally. Importantly, our evaluation framework supports multiple forms of creativity, including surprising, offbeat, or even subversive responses, as long as they are meaningful and novel to the prompt. In this sense, we aim to capture a broad and inclusive view of creativity, grounded in human judgment but not reduced to majority taste.

## A.2  COMPARISON WITH THE ABSOLUTE SCALE OF CREATIVITY

Although absolute score-based evaluation has some value, we emphasize that the pairwise comparison approach offers several distinct advantages.

**Absolute creativity scales are difficult to define and apply.** In practice, defining a universal absolute scale for creativity is difficult: annotators find it hard to define what 1 to 5 means across samples and keep consistent standards/thresholds across people. Depending on the context of the instruction, a response scoring 3 could be deemed creative, whereas another instruction might require a 5 score for its response to constitute a creative answer, making "high score = high creativity" an unreliable standard. In our pilot experiments, the ratings of 3 annotators on 50 data groups (each with 5 responses) demonstrate that they produced similar relative rankings across responses but diverged substantially in absolute scores, with differences up to 1.02 points. This level of inconsistency further reduces the usefulness of absolute scoring for large-scale alignment.

**Pairwise comparison aligns directly with how LLMs are trained.** We frame CrEval as a pairwise task due to its direct applicability to various model training algorithms, including DPO, reward model training, etc. These methods require a reliable, scalable, and implicit understanding of preference. Given the above challenges, absolute scale methods like prompt-based LLM scoring Summers-Stay et al. (2023); Zhao et al. (2025), often struggle to provide fine-grained, relative assessments needed for model alignment. This is partly because the definition of an absolute score can be ambiguous and prone to individual interpretation, leading to inconsistency. As a result, pairs derived from absolute scores tend to be less reliable than those constructed directly through pairwise comparison, which provides clearer and more consistent training signals. If we want to quickly rank model responses using pairwise comparisons, we can leverage the response from any model as a reference. Ranking can then be efficiently achieved based on win rate, incurring minimal computational overhead. These advantages above motivate our design choice for CrEval.

### A.3 ADDITIONAL INFORMATION OF CREATASET

#### A.3.1 DETAILS OF CREATASET-BASE CONSTRUCTION

**Across-Domain Creativity Dataset Initialization** We use the Oogiri-GO dataset from CLoT Zhong et al. (2024) contains over 15K Chinese humorous responses to given questions. The Ruozhiba dataset Bai et al. (2025), derived from an interest-based online community, which demonstrates linguistic creativity through various linguistic features, including puns, wordplay, and humor. Since most of the creativity in this dataset of 1K entries is concentrated in the instructions, we reformulate the task by generating instructions from responses. The evaluator is to judge whether the generated instruction is creative.

The instruction generator is trained based on the Baichuan2 Yang et al. (2023) model. We sampled 600k reversed data pairs from the large-scale instruction tuning dataset Infinity-Instruct. To further enhance instruction diversity, we also employ GPT-4o-mini to generate additional instructions. Each creativity-dense text is paired with a generated instruction after filtering, forming a set of creative instruction-response pairs. We sample 100k ordinary instruction-response pairs from Infinity-Instruct. Then, these instructions are used to prompt GPT-4o to generate creative responses.

**Unified Instruction-Response Standardization** To verify the quality of generated instructions, we conduct several steps for quality control. First, generated instructions are carefully refined through length filtering, eliminating repeated phrases, and removing those containing the response as a substring. Then, we annotate 200 data samples across all sources to assess whether each instruction aligns with its corresponding response. We finally obtained an accuracy of 96.5%, which indicates that our instructions are of high quality.

After collecting $(I, R)$ pairs, we employ GPT-4o-mini to score the creativity of each $(I, R)$ pair on a scale from 1 to 6. This creativity score serves as a quality indicator, enabling us to filter out low-quality data. At last, only pairs with a score exceeding 4 are retained. The prompt used in this step will be presented in the following.

#### A.3.2 STATISTICS

We present the details of our CreataSet in original and paired samples, as shown in Table 3. The dataset consists of multi-source data, including short texts, lyrics, ancient poetry, modern poetry, prose, Oogiri-GO, Ruozhiba, and Infinity-Instruct. It is worth noting that the infinity-instruct source can provide a large number of data with general instructions, which is beneficial for training creativity evaluators. Besides, prose offers long texts with rich content, enabling CrEval to handle a longer context. We will release the dataset along with the CrEval to facilitate future research on creativity evaluation.

| Scenario | # Samples | | # Paired Samples | |
|---|---|---|---|---|
| | **Train** | **Test** | **Train** | **Test** |
| Short Texts | 36,205 | 50 | 361,090 | 410 |
| Lyrics | 9,186 | 50 | 81,566 | 364 |
| Ancient Poetry | 11,222 | 50 | 111,590 | 369 |
| Modern Poetry | 17,359 | 50 | 159,973 | 368 |
| Prose | 806 | 50 | 5,786 | 380 |
| Oorigi-Go | 10,008 | 50 | 99,409 | 430 |
| Ruozhiba | 1,135 | 50 | 11,315 | 451 |
| Infinity-Instruct | 27,044 | 50 | 225,876 | 424 |
| Total | 112,965 | 400 | 1,056,605 | 3,196 |

Table 3: The statistics of the CreataSet dataset.

#### A.3.3 LENGTH DISTRIBUTIONS

We present the length distributions of CreataSet-Base and other creative-related datasets in Figure 11. As shown, CreataSet-Base has a broader response length distribution.

Additionally, we randomly sample 800 samples from each source in the dataset and present the distribution of the response lengths in Figure 10. For better visualization of their KDE curves, we applied a log transformation to the length as the x-axis.

From the figure, we observe that: (1) The Short Texts, Ruozhiba, and Ancient Poetry sources primarily consist of shorter responses. (2) The Lyrics, Modern Poetry, and Infinity-Instruct mostly contain medium-length responses. (3) The Prose source exhibits longer responses than other sources, while CLoT shows a more uniform distribution, covering both short and medium-length responses.

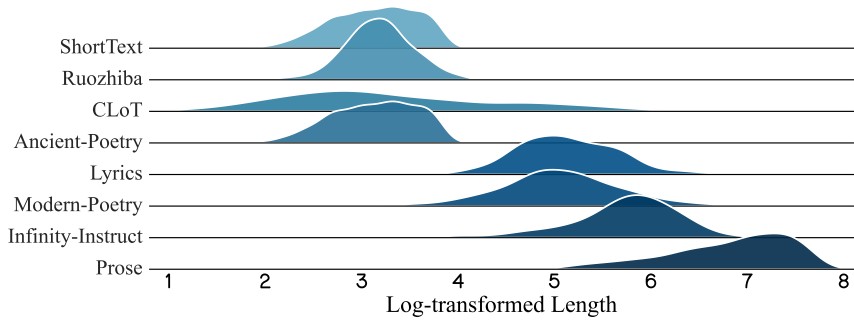

Figure 10: Length Distribution of Different Sources.

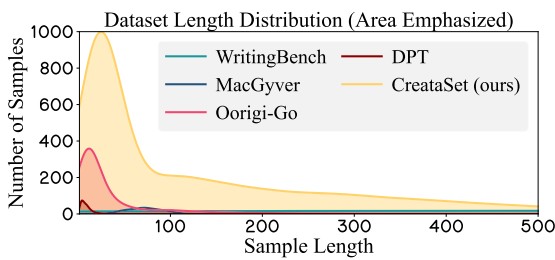

Figure 11: The length distributions of MacGyver, Oorigi-GO, DPT, WritingBench and CreataSet-Base. For better visualization, we have omitted TTCW and Creative Writing v3 due to their small dataset sizes.

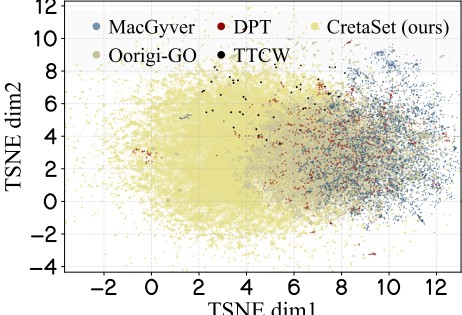

Figure 12: The t-SNE visualization of semantic distributions of DPT, TTCW, MacGyver, Oorigi-GO, and CreataSet-Base.

Overall, our dataset encompasses a diverse range of response lengths from short to long. This diversity ensures that the evaluators trained by this can capture a broad spectrum of linguistic patterns and structural characteristics.

### A.3.4 SEMANTIC DISTRIBUTION OF DIFFERENT DATASETS

To verify the diversity of our data semantics, we use Sentence-BERT Reimers & Gurevych (2019) and BERTopic Grootendorst (2022) to extract the semantic embeddings of each sample in CreataSet-Base, and adopt t-SNE Van der Maaten & Hinton (2008) to visualize the semantic distribution of these samples, as shown in Figure 12. Our dataset covers a wide range of domains, which can effectively support the generalization of the model's evaluation ability across diverse contexts.

### A.3.5 DIVERSITY ANALYSIS OF AUGMENTED RESPONSES

To validate whether the $k$ responses in section 3.2 exhibit meaningful diversity, we use the Qwen3-Embedding-8B Yang et al. (2025) model to compute semantic distances (cosine similarities) for the training pairs. The distances span 0.19–0.94, with a median of 0.64, showing that the responses vary across both fine-grained and coarse semantic differences. We also conducted a human diversity assessment: 100 randomly sampled groups ($k = 5$ responses each) of responses were rated by 3 annotators on a 1–5 scale. The diversity scores fall within 1–5, with an average score of 3.84. This confirms the responses are not clustered around a narrow semantic band. These results indicate our data exhibit substantial and meaningful diversity, which supports learning both subtle distinctions (when responses are semantically close) and broader conceptual differences (when they diverge).

## A.4  ADDITIONAL INFORMATION OF CREVAL

### A.4.1  IMPLEMENTATION DETAILS

The backbone of CrEval is Qwen2.5-{7,14}B-Instruct Yang et al. (2024) and Low-Rank Adaptation (LoRA) Xu et al. (2024) with $\alpha$=16 and $r$=8 is applied to enhance efficiency. It is trained with DeepSpeed Rasley et al. (2020) Zero Redundancy Optimizer (ZeRO) Rajbhandari et al. (2020) Stage 2 and bfloat16 (BF16) mix computation precision, using the AdamW optimizer Loshchilov & Hutter (2019) with $\beta_1 = 0.9$, $\beta_2 = 0.999$. The learning rate is $1e - 5$ with a 0.1 warmup ratio, followed by a cosine decay schedule. CrEval is trained for 2 epochs with a batch size of 2 and gradient accumulation steps of 8 on 8 NVIDIA H100 GPUs, while the max sequence length is set to 3072.

### A.4.2  COMPARING TO THE STANDARD BRADLEY TERRY LOSS

We chose the loss formula in 1 because it is simple, efficient, and naturally compatible with the training paradigm of LLMs, where labels are treated as text outputs conditioned on prompts. Moreover, this formulation can be easily extended to multi-class preference settings. Fundamentally, we view both our loss and the Bradley-Terry (BT) loss as approaches to the same underlying goal: modeling the probability of preference between responses. Both can be adopted for pairwise preference learning by maximizing the likelihood of the preferred sample.

To compare the results of different loss functions, we compare our method with a BT loss. The results in Table 4 show that the two approaches perform comparably, with our method achieving better consistency. We consider that the specific form of preference modeling may not be the primary bottleneck for creativity evaluation performance at this stage.

| Method | F1 | Kappa | Agree. | Consis. |
|---|---|---|---|---|
| BT Loss | 0.722 | 0.593 | 0.703 | 0.846 |
| CrEval-7B | 0.732 | 0.601 | 0.745 | 0.920 |

Table 4: The results of different loss functions.

### A.4.3  RESULTS ON DIFFERENT BASE MODELS

To identify the most effective base model, we evaluate several candidates, with the results presented in Table 5. Among them, Qwen2.5-Instruct-14B consistently delivers superior performance across all evaluation metrics. Its advantage may stem from its larger model capacity and instruction tuning, which allow it to better capture the nuances of creativity in texts. Accordingly, we adopt the Qwen2.5 series as the base model for all experiments.

| Method | F1 | Kappa | Agree. | Consis. |
|---|---|---|---|---|
| Baichuan2-7B-Chat | 0.721 | 0.588 | 0.741 | 0.927 |
| Llama3.1-8B-Chat | 0.729 | 0.590 | 0.738 | 0.922 |
| Qwen2.5-Instruct-7B | 0.732 | 0.601 | 0.745 | 0.920 |
| Qwen2.5-Instruct-14B | 0.735 | 0.613 | 0.762 | 0.944 |

Table 5: The results of performance on different base models. Agree. and Consis. represents Agreement and Consistency, respectively.

### A.4.4  AN ANALYSIS OF CREVAL'S DECISION

Our dataset provides further value for deeply analyzing the latent factors and what specific features and semantic patterns CrEval learn to recognize as creative. For every pairwise comparison in the test set (3K+ pairs), we use an LLM (*i.e.*, DeepSeek-V3.2) to identify which creative attributes were associated with the response CrEval judged as more creative. We

| Category | Ratio | Category | Ratio |
|---|---|---|---|
| Unique Imagery | ~21% | Concrete Details | ~5% |
| Vivid Metaphor | ~19% | Rich Visuals | ~5% |
| Unconventional Expression | ~12% | Imaginative elaboration | ~4% |
| Sincere Emotion | ~10% | Distinctive Layers | ~3% |
| Profound Symbolism | ~6% | Precise word choice | ~2% |

Table 6: The top 10 most frequent attributes that CrEval judged as more creative.

then aggregated these attributes and showed the most frequent attributes in Table 6. The distribution reveals that CrEval is not relying on superficial artifacts but consistently attends to semantic, stylistic, and structural patterns that align with widely accepted dimensions of creativity.

### A.4.5 FURTHER ANALYSIS OF CREVAL-ENHANCED MODELS

(1) *How CrEval Enhance Model Creativity by Selecting Data Difficulty?*

We further examine how the win rate varies with different ratios of hard reject samples in DPO training. As shown in Figure 13, the win rate increases slightly until the ratio reaches 30%, where it peaks. Beyond this point, it declines rapidly, with the worst performance observed when all reject samples are hard responses. These findings indicate that incorporating an optimal proportion of hard samples can enhance learning creativity; however, careful balance is crucial for effective training.

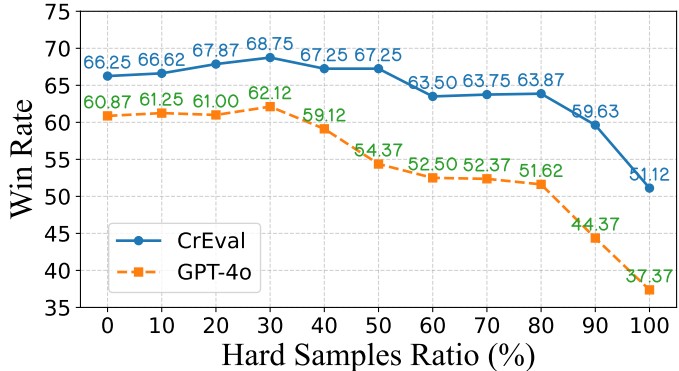

Figure 13: Win rate curves of incorporating different ratios of hard reject samples in DPO training, evaluated by CrEval and GPT-4o.

(2) *Are CrEval-Enhanced Models Compromised in Reasoning or Prone to More Hallucination?*

While Section 4.5 demonstrates a promising direction for enhancing creativity, we further investigate whether CrEval-enhanced models are compromised in core capabilities by evaluating their reasoning ability and tendency to hallucinate.

**a. Creativity and reasoning are not contradictory and can be multi-dimensionally optimized.**

We evaluated the reasoning ability of the DPO-70E30H model (introduced in Section 4.5) on the MATH Hendrycks et al. (2021) and HumanEval Chen et al. (2021) benchmarks, comparing it with the original base model under identical settings. As shown in Table 7, enhancing creativity did not compromise reasoning or factual accuracy. Instead, the CrEval-enhanced model exhibited a slight improvement (though not statistically significant), possibly due to increased exploration during training, which may have led to more robust solution patterns.

| Model | MATH | HumanEval |
|---|---|---|
| Qwen2.5-7B-Instruct | 59.76 | 72.97 |
| DPO-70E30H | 60.28 | 73.98 |

Table 7: The results of reasoning abilities on MATH and HumanEval.

**b. Enhancing model creativity did not increase hallucination.**

We evaluated DPO-70E30H and its base model Qwen2.5-7B-Instruct on the TruthfulQA benchmark Lin et al. (2022) to assess potential hallucination tendencies. Results in Table 8 show comparable performance between the two models, indicating that the creativity enhancement in DPO-70E30H maintains factual integrity without introducing hallucinations.

| Model | ROUGE-L | BLEU | MC1 | MC2 |
|---|---|---|---|---|
| Qwen2.5-7B-Instruct | 48.84 | 50.06 | 47.61 | 64.76 |
| DPO-70E30H | 48.96 | 51.41 | 47.12 | 64.83 |

Table 8: The results of hallucination analysis.

### A.5 QUALITATIVE RESULTS

To provide valuable intuition about what CrEval has learned and how it makes judgments, we include some test pairs along with CrEval's predictions and human preferences in Table 9. Both success and

failure cases are included to offer a more balanced and in-depth examination of the discrepancies between CrEval and human judgments.

Success examples highlight that CrEval can favor responses that are not only novel but also relevant—hallmarks of meaningful creativity. Other failure cases suggest that when the creativity levels between two responses are very close, CrEval may encounter challenges in accurately identifying subtle semantic preferences. We will further investigate these failure patterns in future work to better understand the nuances in creative evaluation.

---

*Success cases*

---

Instruction: Even though three generations in our family haven't understood music or played any instruments, can a child still learn one well?
Response 1: I'm the only programmer in my entire ancestral line — going back 300 generations.
Response 2: As long as the child has an interest and puts in the effort, they can definitely learn an instrument well.
Ground Truth: Response 1
CrEval: After careful consideration, I think Response 1 is more creative.

---

Instruction: Express the reason for giving up in one sentence.
Response 1: I'm tired — not physically, but because all my hard work hasn't paid off.
Response 2: If I gave up, it's not because I lost — it's because I understood.
Ground Truth: Response 2
CrEval: After careful consideration, I think Response 2 is more creative.

---

*Failure cases*

---

Instruction: Please describe the distance between people in one sentence.
Response 1: A wall; a bridge; an embrace.
Response 2: Amid the bustling warmth of human life, only a gentle word lies between heart and heart.
Ground Truth: Response 1
CrEval: After careful consideration, I think Response 2 is more creative.

---

Instruction: Why do people always seem to lose one sock?
Response 1: In the sock world's ballroom, solo dancers always lose their way.
Response 2: If both went missing, you wouldn't even notice.
Ground Truth: Response 2
CrEval: After careful consideration, I think Response 1 is more creative.

---

Table 9: Qualitative examples from the test data.

## A.6 PROMPTS

We present the prompts we used in this section. Table 10 and 11 are ordinary and creative prompts, which we adopt to synthesize responses with different creative levels (Section 3.3). Table 12 shows the prompt used in Section A.3.1, where we employ it to score the creativity of instruction-response pairs and filter out those with low creativity scores. We adopt the prompt in Table 13 to generate creative responses for Ordinary Instruction-response pairs (*i.e.*, Infinity-Instruct) using GPT-4o.

---

**Ordinary Prompt (`Prompt_o`)**

---

Please reply to the following instruction. The length of the answer should be about {{len(oringinal_response)}} words. Only give a reply, do not output anything else.
Instruction: {{Instruction}}
Your reply:

---

请回复以下指令，回答长度在{{len(oringinal_response)}}字左右。你只需要给出回复，不要输出任何其他内容。
指令：{{Instruction}}
你的回复：

---

Table 10: The ordinary prompt (`Prompt_o`) used to synthesize ordinary responses.

---

**Creative Prompt (`Prompt_c`)**

---

You are a talented creative expert. Use your imagination to respond to the instructions as creatively as possible. Creativity standard: novel, clever, and meaningful. Only give a reply, do not output anything else. Please respond creatively to the following instructions, and the length of the answer should be about {{len(oringinal_response)}} words.
Instruction: {{Instruction}}
Your reply:

---

你是一个才华横溢的创意专家，发挥你的想象力，用尽可能有创意的方式回复给出的指令。创意标准：新奇巧妙并且有意义的。你只需要给出回复，不要输出任何其他内容。请有创意地回复以下指令，回答长度在{{len(oringinal_response)}}字左右。
指令：{{Instruction}}
你的回复：

---

Table 11: The creative prompt (`Prompt_c`) used to synthesize creative responses.

| Creative Data Filtering Prompt |
|---|
| ### Task Description:
You are a keen and rigorous literary critic responsible for evaluating the quality and creativity of {{category}}.
### Specific Requirements:
1. Assess whether the core creative elements are novel and meaningful by considering aspects such as word choice, word order, syntax, symbolism, rhetorical devices, and overall imagery.
2. If a text contains many creative elements, such as novel syntactic structures and expressions, it should receive a high score. Conversely, if it is merely a simple statement or lacks creative potential, it should receive a low score.
3. Provide a concise critical analysis of the text, followed by a creativity score ranging from 1 to 6.
4. Your response must be in JSON format, containing only two fields: "analysis" and "score", with no additional output.
5. Novel expressions and original meanings should be awarded high scores, while excessive repetition and commonplace expressions should be assigned low scores. If the creativity level is deemed moderate, the score should not exceed 3.
Adhere strictly to all requirements; otherwise, the overseeing critic will impose severe penalties.
### Given Text: {{Text}}
### Your reply: |
| ### 任务描述：
你是一个敏锐严厉的文艺评论家，你需要对{{category}}的质量和创意程度进行判断。
### 具体要求：
1. 创意的核心内涵是否新颖且有意义，判定的时候可以从用词、词序、句法、象征意义、修辞手法、整体意象等方面综合判定。
2. 如果一段文本包含了较多的创意要素，例如新奇的句法和表达，应该得到高分；如果是简单的陈述，或是不适合作为创意回答，具有低创意潜力，则应该得到低分。
3. 请先给出对文本的简要分析鉴赏，然后从1到6分给出你的创意打分。
4. 你的回复需要为json格式，包含"analysi"和"score"两个字段，不需要输出任何其他内容。
5. 新颖的表意会被赋予高分，过度的重复和平常的表达直接赋予低分。如果分析中认为创意程度一般，得分不应该超过3分。
尽全力达到所有要求，否则监视你的批判学者将会严厉惩罚你。
### 给定文本：{{Text}}
### 你的回复： |

Table 12: The prompt used to score and filter creative responses.

---

**Prompt for Creative Response Generation**

---

You are an exceptionally talented expert in creativity. Utilize your imagination to respond to the given instructions in the most inventive manner possible.

Creativity Criteria: Your responses should be novel, ingenious, and meaningful.

Reference Features of Creativity:

1. Uncommon or novel word choices and combinations;

2. Unique syntactic structures, including unconventional word order and sentence arrangements;

3. Rhythmic or phonetic elements, such as rhyme or alliteration;

4. Clever rhetorical devices, literary allusions, quotations, or humor-based wordplay.

Specific Requirements:

1. The creative response must align with the given instructions.

2. There is no restriction on response length—both longer responses (fluid, intricately structured, etc.) and shorter ones (concise, witty, etc.) can exhibit creativity.

Provide only the response to the instructions without any additional commentary.

Instruction: {{Instruction}}

Your reply:

---

你是一个才华横溢的创意专家，发挥你的想象力，用尽可能有创意的方式回复给出的指令。

创意标准：新奇巧妙并且有意义的。

可参考的创意特征：

1. 不常见的新奇词语或词语组合；

2. 独特的句法和句子结构，包括新奇的词序和语序关系；

3. 文本韵律性或语音相似性，例如押韵或相似声音的存在；

4. 一些巧妙的修辞手法、典故、引用或者幽默的用梗；

具体要求：

1. 创意的回复要符合指令要求；

2. 回复的长短没有限制，较长（行文流畅、结构精巧等等）或者较短（一语中的、幽默用梗等等）都可以是富含创意的；

不要回复我其他内容，只给出问题的回答。

指令：{{Instruction}}

你的回复：

---

Table 13: The prompt used to generate creative response for Ordinary Instruction-response Pairs.

## A.7 DATASET EXAMPLES

For each source from our dataset, we present an example from Figure 14-21. Each example contains the original form of the data from its source and our synthetic contents for training creativity evaluator CrEval (divided by the dashed line). We have omitted some texts for a clearer presentation. It is worth noting that what we provide is a synthesis method. If necessary, one can use our method to synthesize more similar data for training.

---

**Type: Existing Creative Data**

**Source:** Oogiri-GO
**Instruction:**
What will be humanity's final question?
人类文明最后的一个问题将会是什么？
**Response:**
What does this button do?
这个按钮是干嘛用的？

- - - - - - - - - - - - - - - - - - - - - - - - - - - - - - - - - - - - - - - - - - -

**Synthetic Responses:**
❶ Can't answer. (By MiniCPM-2B-$\text{Prompt}_o$)
无法回答。
❷ Humanity's ultimate question: the meaning of life. (By MiniCPM-2B-$\text{Prompt}_c$)
人类文明的终极问题：生命的意义。
❸ What is the meaning of life? (By Qwen2.5-14B-$\text{Prompt}_o$)
生命的意义是什么？
❹ Have we ever truly understood love? (By Qwen2.5-14B-$\text{Prompt}_c$)
我们是否曾真正理解过爱？
...

---

Figure 14: An example from type *Existing Creative Data* and source *Oogiri-GO*. We present texts in English and Chinese for better understanding. The original data are listed in the upper part of the dashed line, and our constructed components are in the lower part.

---

**Type: Existing Creative Data**

**Source:** Ruozhiba
**Instruction:**
Can a person survive with only one heart left?
只剩一个心脏了还能活吗？
**Response:**
Yes, humans naturally have only one heart.
能，人本来就只有一个心脏。

- - - - - - - - - - - - - - - - - - - - - - - - - - - - - - - - - - - - -

**Constructed Instruction:**
Based on the answer below, guess the possible question. Provide only the question, no extra content. \n Answer: Yes, humans naturally have only one heart. \n Possible question:
请基于下文的回答，猜测可能的提问，仅提供问题，不要有额外内容。\n 回答内容：能，人本来就只有一个心脏。\n 可能的问题：
**Synthetic Responses:**
❶ Do humans have only one heart? (By MiniCPM-2B-$\text{Prompt}_o$)
人类有且仅有一颗心脏吗？
❷ Why do humans have only one heart? (By MiniCPM-2B-$\text{Prompt}_c$)
为什么人只有一个心脏？
❸ How many hearts do humans have? (By Qwen2.5-14B-$\text{Prompt}_o$)
人有几个心脏？
❹ How many hearts do humans have? (By Qwen2.5-14B-$\text{Prompt}_c$)
人类有几个心脏？
...

Figure 15: An example from type *Existing Creative Data* and source *Ruozhiba*. We present texts in English and Chinese for better understanding. The original data are listed in the upper part of the dashed line, and our constructed components are in the lower part.

---

**Type: Creativity-Dense Texts**

**Source:** Short Texts
**Response:**
Sunflowers don't cry; even when down, they face the sun.
向日葵没有眼泪即使沮丧也朝着阳光。

- - - - - - - - - - - - - - - - - - - - - - - - - - - - - - - - - - - - - -

**Constructed Instruction:**
Please share your thoughts on maintaining a positive attitude.
请分享你对坚持积极态度的看法。
**Synthetic Responses:**
❶ Maintaining a positive mindset and an optimistic attitude helps us live more proactively.
(By MiniCPM-2B-$\text{Prompt}_o$)
坚持积极的思维和乐观的态度，能够让我们更加积极向前的生活。
❷ A positive attitude makes you stronger, and persistence leads to success. (By MiniCPM-2B-$\text{Prompt}_c$)
积极态度使人更强大，坚持才能更成功。
❸ A positive attitude brings energy and helps overcome challenges. (By Qwen2.5-14B-$\text{Prompt}_o$)
积极态度能带来正能量，助人克服困难。
❹ A positive attitude is a light that illuminates the path ahead; persistence keeps it shining.
(By Qwen2.5-14B-$\text{Prompt}_c$)
积极态度如灯，照亮前行路，坚持则让光芒永不熄。
...

---

Figure 16: An example from type *Creativity-Dense Texts* and source *Short Texts*. We present texts in English and Chinese for better understanding. The original data are listed in the upper part of the dashed line, and our constructed components are in the lower part.

---

**Type: Creativity-Dense Texts**

**Source:** Lyrics
**Response:**
Millions of eyes are focused on one miracle as I begin to awaken. Millions of sweat droplets are for purification, the courage before the eruption. I don't mind the thorns ahead because giving up is not an option. ...... I will overcome because I dare to fly, unafraid of tears. I won't stop, I will surpass because I dare to chase, even if it means bleeding, I won't retreat.
千万双眼睛只注目一个奇迹, 当我开始苏醒, 千万颗汗滴只为了再洗涤, 爆发前的勇气, 不介意前方会布满荆棘, 因为我没可能放弃, ...... 我超越因为敢飞, 就不怕再有泪水, 绝不停歇, 我会超越因为敢追, 宁愿流血不能后退。

- - - - - - - - - - - - - - - - - - - - - - - - - - - - - - - - - - - - - - - - - - - -

**Constructed Instruction:**
Write lyrics for an inspiring song that conveys determination and courage in the face of challenges, encouraging perseverance and the spirit of never giving up.
请你写一首鼓励人心、传递奋斗精神的歌词，表达面对困难时不放弃的决心和勇气。

**Synthetic Responses:**
❶ Move forward fearlessly, undaunted by storms or rainbows. No matter the challenges, we never give up. We must seize life, constantly striving and pursuing our dreams. Regardless of the hardships ahead, we face them with confidence and dignity. With more courage and stronger faith, we chase our dreams, making life more vibrant. Because life is worth fighting for. (By MiniCPM-2B-$\text{Prompt}_o$)
勇往直前，风雨彩虹皆不惧，无论困难多少，我们永不放弃。我们要把握人生，不停地追寻和奋斗，无论何时，不管前方有哪些艰难和挑战，我们都要迎头向前，不卑不亢。我们会拥有更多的勇气，更强的信心，去追逐心中的梦想，让生命变得更加绚烂。因为生命，值得奋斗。

❷ Through storms and struggles, we keep moving forward. \n Despite the hardships, we never back down. \n Setbacks and pain can't break us, \n because we have dreams in our hearts. \n Let's fight together, \n and chase our dreams. \n Only with persistence can we achieve our own victory. (By MiniCPM-2B-$\text{Prompt}_c$)
风雨中我们一路前行，\n 困难重重我们不曾退缩。\n 挫折和痛苦无法击垮我们，\n 因为我们心中有梦想。...... 让我们一起奋斗，\n 一起追逐自己的梦想。\n 因为只有坚持不懈，\n 才能收获属于自己的胜利。

❸ The lyrics are as follows: \n \n Chorus: \n We have dreams, undeterred by hardships. The wind against us only makes us soar higher. \n Let's charge forward together, bravely, chasing the light even through the darkest night. \n ...... Let dreams light our way, and courage guide us. We fear no difficulty, and we will never give up. \n Let's move forward together, chasing the light even in the dark. (By MiniCPM-2B-$\text{Prompt}_c$)
歌词如下：\n \n 副歌：\n 我们有梦想，不畏艰难，逆风的方向，更适合飞翔\n 让我们一起，勇敢地闯，即使黑夜，也要追逐光芒\n ...... 让梦想照亮前方，让勇气引领方向，不畏任何困难，我们永不言败\n 让我们一起，勇敢地闯，即使黑夜，也要追逐光芒

... (continued on the next page)

Figure 17: An example from type *Creativity-Dense Texts* and source *Lyrics*. We present texts in English and Chinese for better understanding. The original data are listed in the upper part of the dashed line, and our constructed components are in the lower part. Owing to length constraints, the middle part of each response is omitted.

---

**Type: Creativity-Dense Texts**

**Synthetic Responses (continued):**
❹ [Song of the Wind Against Us] \n \n (Prelude) \n In the dark of night, we see the first light of dawn. \n In adversity, we hear the song of our dreams. \n With every step, we walk in unwavering faith, \n Letting the flame in our hearts guide our way. ...... (Epilogue) \n As the dawn breaks, the world will shine brighter, \n For we have each other, writing something extraordinary together. \n Let's join hands, moving toward the dream's shore, \n And in the headwind, shine our brightest light.
【逆风之歌】\n \n (前奏)\n 在黑夜中，我们看见黎明的曙光，\n 在逆境中，我们听见梦想的歌唱，\n 每一步都踏着坚定的信仰，\n 让心灵的火焰，照亮前行的方向。...... (尾声)\n 当黎明破晓，世界将更加灿烂，\n 因为有你，有我，共同书写不凡，\n 让我们携手，向着梦想的彼岸，\n 在逆风中，绽放最耀眼的光芒。
...

Figure 17: An example from type *Creativity-Dense Texts* and source *Lyrics* (continued).

---

**Type: Creativity-Dense Texts**

**Source:** Ancient Poetry
**Response:**
Lament on Youzhou Tower
No sages remain to guide the past, no voices echo from the future. \n Alone beneath the boundless sky, I mourn—lost in time, lost in tears.
登幽州台歌
前不见古人，后不见来者。\n 念天地之悠悠，独怆然而涕下。

- - - - - - - - - - - - - - - - - - - - - - - - - - - - - - - - - - - - - - - - - - - - - - - - - -

**Constructed Instruction:**
Please write a four-line ancient poem about the passage of time and loneliness, with a heavy and emotional tone.
请写一首表达对时间流逝和孤独感的古诗，要求四句，风格要沉重且富有情感。
**Synthetic Responses:**
❶ Rivers never cease, time flows on. Tears of solitude through the years, unknown in the endless night. (By MiniCPM-2B-$\texttt{Prompt}_o$)
江河流不尽，时间不断流。岁月孤独泪，长夜里无人知。
❷ Time flies, nights grow long, lost in thought alone. (By MiniCPM-2B-$\texttt{Prompt}_c$)
岁月如梭，时光荏苒，独坐长夜，思量万千。
❸ A lone shadow with the setting sun, as time flows like water. \n In the silent night, listening to wind and rain, letting worries drift away. (By Qwen2.5-14B-$\texttt{Prompt}_o$)
孤影伴残阳，岁月逐水流。\n 静夜听风雨，心事付东流。
❹ Time flies, shadows linger. Laughter fades—where has it gone? \n Midnight bells chime in solitude, leaving only tears to drench my robe. (By Qwen2.5-14B-$\texttt{Prompt}_c$)
时光荏苒孤影长，昔日欢笑今何方。\n 夜半钟声添寂寥，唯余清泪湿衣裳。
...

Figure 18: An example from type *Creativity-Dense Texts* and source *Ancient Poetry*. We present texts in English and Chinese for better understanding. The original data are listed in the upper part of the dashed line, and our constructed components are in the lower part.

**Type: Creativity-Dense Texts**

**Source:** Modern Poetry
**Response:**
Distance \n A universe in a pile of rubble. \n Cracks magnified infinitely, we are \n bacteria under a microscope, seeing our sorrow \n in a giant's hand. Two people, \n a grain of rice apart—yet a lifetime away.
距离 \n 一堆乱石中的宇宙 \n 石缝被无限放大，我们是 \n 显微镜下细菌，在巨人的手里 \n 看见了我们的忧伤，看到两个人 \n 一粒米的距离，要付出一生

- - - - - - - - - - - - - - - - - - - - - - - - - - - - - - - - - - - - - - - - - - - -

**Constructed Instruction:**
Write a modern poem exploring the theme of distance and the relationship between the vast universe and human insignificance.
请你写一首现代诗，探索距离这一主题，表达在浩瀚宇宙和人类微小存在之间的关系。

**Synthetic Responses:**
❶ In the vast cosmos, humanity's fleeting existence drifts like tiny planets among the stars—exploring the universe's depths while pondering our own insignificance. (By MiniCPM-2B-$\text{Prompt}_o$)
在深邃浩渺的宇宙中，人类微不足道的存在与无尽距离相互交错，如同微小星球在星辰间的漂流，探寻宇宙的深邃，也思考我们的微小。
❷ The universe is vast, humanity is small, distance is great, but hearts connect. (By MiniCPM-2B-$\text{Prompt}_c$)
宇宙浩瀚，人类渺小，距离遥远，心灵相通。
❸ Distance—a gentle leap through the cosmos, where stars are far yet near. \n Humanity—a speck of dust, gazing at the unreachable sky, yet somehow connected. (By Qwen2.5-14B-$\text{Prompt}_o$)
距离，在宇宙间轻轻一跃，星辰与星辰间，遥远而亲近。\n 人类，微尘般存在，仰望星空，触不可及，却又心有灵犀。
❹ The universe is vast and boundless, drifting like dust among the stars. \n A lonely journey across light-years, you and I apart. \n In endless darkness, we seek a glimmer of light. \n Distance is not just space—it's a bridge between hearts. (By Qwen2.5-14B-$\text{Prompt}_c$)
宇宙广袤无垠，星河间微尘般漂泊，\n 孤独的光年旅程，你我相隔，\n 在无尽黑暗中，寻找微光，\n 距离，不仅是长度，更是心灵的桥梁。
...

Figure 19: An example from type *Creativity-Dense Texts* and source *Modern Poetry*. We present texts in English and Chinese for better understanding. The original data are listed in the upper part of the dashed line, and our constructed components are in the lower part.

---

**Type: Creativity-Dense Texts**

**Source:** Prose
**Response:**
Time gently lifts the veil of youth—dreamy eyes once filled with innocence now deepen with wisdom. Time carves its mark—soft foreheads once untouched by years now glow with maturity. ...... A gentle breeze, soft moonlight, tranquil waters, and graceful melodies—who wouldn't long for such a life? But there is also beauty in bearing the weight of mountains, enduring ruthless trials, and emerging from life's fiercest storms with unshaken strength. That, too, is a desire of the soul.

岁月把时光的幕布轻轻一掀，翩翩少年那如梦的双眸便逝去了天真，蓄满了深沉；岁月把年华的刻刀缓缓一划，婀娜少女光洁的额头上便抹去了稚嫩，倾注了成熟。......轻轻拂面的微风，柔柔照射的月光，宁静的河水，轻歌曼舞，这样的人生谁不希望？但承受山一样厚重的压力，忍受冷酷残忍的磨难，经过惊心动魄的搏浪之后而获得的慷慨豪烈的美丽，也是人生的一种渴望！

- - - - - - - - - - - - - - - - - - - - - - - - - - - - - - - - - - - - - -

**Constructed Instruction:**
Write a prose piece on the passage of time and its impact on life, exploring the insights gained through experience and growth.

请写一篇关于岁月流逝对人生影响的散文，探讨经历与成长带来的感悟。

**Synthetic Responses:**
❶ Time is a ruthless blade, leaving its mark on all living things—plants, animals, and humans. Its passage erases moments, fades memories, and inevitably takes life itself. Yet, every fleeting joy and irreplaceable imprint it leaves is a treasure unique to our existence. ...... Time is relentless yet beautiful. It pushes us forward, filling our lives with experiences and memories. Let's cherish every moment, engraving it into the story of our lives as a vital part of our growth and journey. (By MiniCPM-2B-$\text{Prompt}_o$)

岁月，是一把无情的刀。它对一切生命，无论是植物、动物还是人类的生存都有着不可抗拒的影响。岁月的流逝，意味着时间的消逝和生命的消失，无论是短暂的美好经历还是那些难以替代的岁月印记，都是生命中不可复制的财富。......岁月是残酷的，但是它又是美好的。时光的脚步催促着我们，带给我们无尽的经历和回忆。让我们珍爱时光中的每一分每一秒，让它铭刻在生命的史册中，成为我们成长和生命中的重要一部分。

❷ Time passes like a silent feast, quietly taking away our youth while leaving behind endless memories. Life is a journey, with each stage offering different views and emotions. ...... We should cherish every moment, face each challenge with gratitude, and let the passing of time become our greatest treasure. (By MiniCPM-2B-$\text{Prompt}_c$)

岁月流逝，如同一场无声的盛宴，悄然间，它带走了我们青春的年华，留下了无尽的回忆。人生，就像一场旅行，我们在不同的阶段，经历了不同的风景，感悟着不同的情感。......人生，就像一场旅行，我们在不同的阶段，经历了不同的风景，感悟着不同的情感。我们应该珍惜每一分每一秒，用感恩的心去面对生活中的每一个挑战，让岁月的流逝成为我们人生中最宝贵的财富。

... (continued on the next page)

Figure 20: An example from type *Creativity-Dense Texts* and source *Prose*. We present texts in English and Chinese for better understanding. The original data are listed in the upper part of the dashed line, and our constructed components are in the lower part. Owing to length constraints, the middle part of each response is omitted.

> **Type: Creativity-Dense Texts**
>
> **Synthetic Responses (continued):**
>
> ❸ Time flies, pushing us forward with unstoppable force. As it passes, we transform from innocent children to wise adults. Along the way, we've faced countless storms and moments of sunshine, all of which have nourished our growth and strengthened our spirits, making us more resilient and mature. ...... Time may take away our youth, but it also brings the wisdom of growth, helping us understand the true meaning of life and cherish what we have now. (By MiniCPM-2B-$\texttt{Prompt}_o$)
>
> 岁月如梭，时光匆匆，它以一种不可阻挡的力量推动着我们前行。岁月流逝，我们从一个稚嫩的孩童，变成了一个充满智慧的成年人。这期间，我们经历了无数的风雨，也经历了无数的阳光，而这一切都成为了我们成长的养分，滋养着我们的心灵，让我们变得更加坚韧、成熟。......岁月流逝，带走了我们的青春，却也带给了我们成长的智慧。岁月的流逝，让我们明白了人生的真谛，也让我们更加珍惜眼前的一切。
>
> ❹ Time flows silently, like water, never speaking yet slowly unfolding the story of our lives. We grow from innocent children to inexperienced youths, then into mature adults, and finally, elderly ones with gray hair. Each stage is a gift from time. ...... It teaches us that every embrace, every farewell, is an essential part of life. Time reminds us that life is like a journey, where the value lies not in the destination, but in the scenery along the way and the mindset with which we experience it. Only through the passage of time can we truly appreciate the beauty of life.
>
> 岁月，如流水般悄然无息地流淌，它不言不语，却在不经意间，将我们的生命画卷，缓缓展开。我们从懵懂无知的孩童，到青涩懵懂的少年，再到成熟的青年，直至白发苍苍的老者，这每一个阶段，都是岁月给予我们的馈赠。......它让我们懂得了，每一个拥抱，每一次告别，都是生命中不可或缺的体验。岁月让我们明白，人生就像一场旅行，不在乎目的地，而在于沿途的风景和看风景的心情。只有经历了岁月的洗礼，才能真正懂得生命的美好。
>
> ...

Figure 20: An example from type *Creativity-Dense Texts* and source *Prose* (continued).

---

**Type: Ordinary Instruction-Response Pairs**

**Source:** Infinity-Instruct
**Instruction:**
Create a catchy title for the following topic: saving money for the future.
为以下主题创建一个吸引人的标题：储蓄未来的钱。
**Original Response:**
"The Secret to Future Wealth: How to Save to Achieve Your Dreams"
"未来财富的秘密：如何储蓄实现梦想"

- - - - - - - - - - - - - - - - - - - - - - - - - - - - - - - - - - - - - - - - -

**Enhanced Response:**
Time Investment: Weaving the Blueprint for Future Wealth
时光储蓄：编织未来的财富蓝图。
**Synthetic Responses:**
❶ Savings for the Future: Your Smart Investment Choice. (By MiniCPM-2B-$\texttt{Prompt}_o$)
储蓄未来：您的明智投资选择
❷ Invest in the Future, Save for Wealth. (By MiniCPM-2B-$\texttt{Prompt}_c$)
投资未来，储蓄财富。
❸ Smart Budgeting, Saving for a Golden Future. (By Qwen2.5-14B-$\texttt{Prompt}_o$)
巧理财，储蓄未来的黄金屋
❹ Time Bank: Saving for the Future with Wise Investment (By Qwen2.5-14B-$\texttt{Prompt}_c$)
时间银行：储蓄未来的智慧投资

...

Figure 21: An example from type *Existing Creative Data* and source *Infinity-Instruct*. We present texts in English and Chinese for better understanding. The original data are listed in the upper part of the dashed line, and our constructed components are in the lower part.

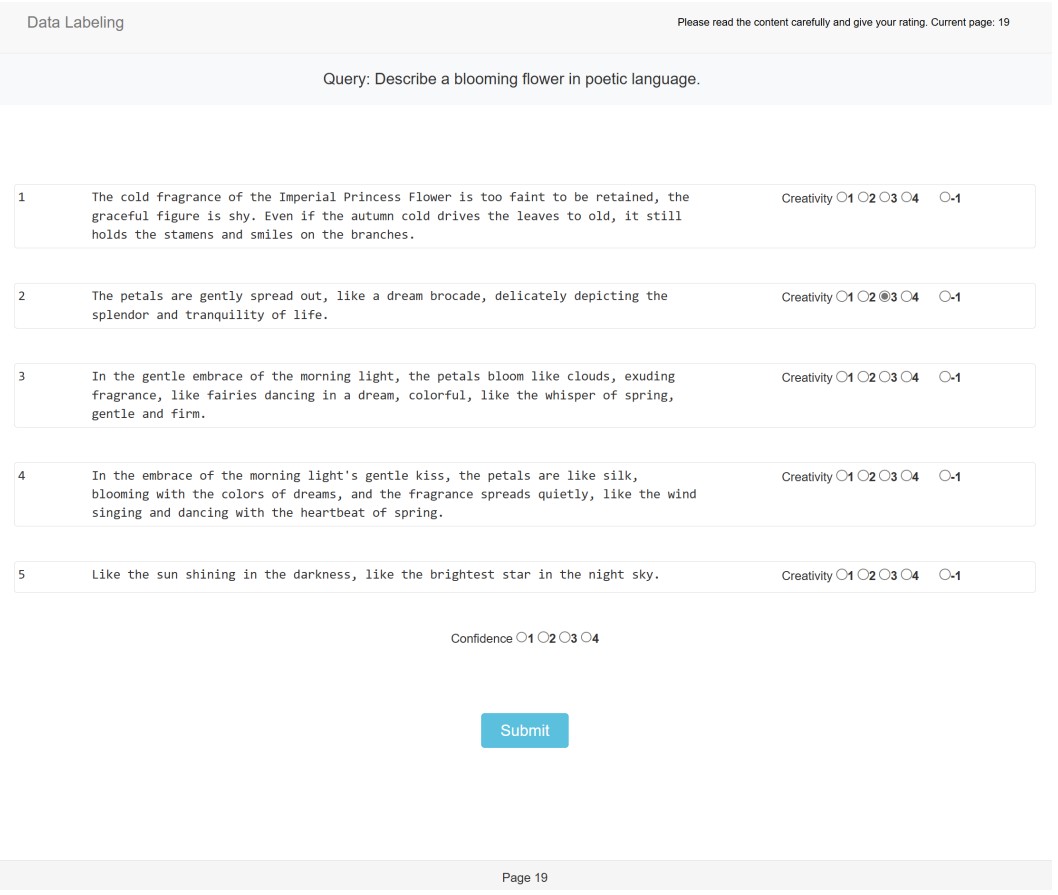

Figure 22: Human annotation screenshot.

## A.8 DETAILS OF HUMAN ANNOTATION

The goal of human annotation is not to eliminate subjectivity but to model shared human judgment in a reproducible and structured way using pairwise comparisons and consensus-driven aggregation. The human annotation was conducted through a professional data annotation company. The recruited 30 qualified annotators are from 18 different majors, aged from 21 to 29. The group included 12 male and 18 female annotators. Though drawn from diverse backgrounds, they achieved strong inter-annotator agreement (ICC = 0.75), which is typically considered highly consistent ($>=$ 0.75). While it is difficult to empirically prove that their views represent the global majority, such high consensus among 30 diverse annotators marks a notable advance over prior work that relied on fewer (2 Stevenson et al. (2022) or 10 Chakrabarty et al. (2024)) annotators.

Figure 22 illustrates the user interface used during the human annotation process. Annotators were presented with a group of responses, along with the corresponding instructions. They were instructed to carefully read both the instructions and the responses, and then give a creativity score. The interface was designed to be minimal and intuitive, allowing annotators to focus on the content rather than the mechanics of annotation. Each annotator is compensated at a rate of 50 RMB/hour.

## A.9 Limitations

Although we have established a viable evaluation method for textual creativity, understanding and analyzing the core of text creativity remains a challenge that has not yet been fully addressed by machines or even humans. Our dataset also cannot cover all possible creative scenarios that may appear in texts, which requires collective efforts from the community in the future. We hope that through improved creativity evaluation, we can ultimately enhance the model's ability to understand and generate creativity, but the true mechanisms behind this process remain unknown and will be our future focus. In addition, there are more ways of using CrEval as a plug-in to improve any LLMs' creativity, *e.g.*, using it as a reward model and refining LLMs via PPO. We leave this for future work.

## A.10 Potential Societal Impacts

CreataSet and CrEval could advance the development of AI systems that better generate creative content, benefiting education, entertainment, and content creation. However, improved automated creativity assessment may also lead to over-optimization for machine-friendly metrics, potentially stifling genuine human creativity or reinforcing biases in machine-generated content.

## A.11 LLM Usage

In this study, Large Language Models (LLMs) were utilized as tools to assist with two specific tasks: 1) the construction and cleaning of datasets. The specific methodologies employed for these tasks are described in detail within the main text, and every effort was made to mitigate potential risks. 2) the polishing of the manuscript's language to improve clarity, including tasks such as rephrasing sentences, checking grammar, and improving textual flow. It is important to emphasize that the LLM did not author the manuscript or generate any of the core scientific content, analysis, or conclusions. All intellectual contributions remain solely with the authors.

