# OpenReview forum: "Evaluating Text Creativity across Diverse Domains: a Dataset and Large Language Model Evaluator"
_ICLR.cc/2026/Conference — ICLR 2026 Poster_

### Official Review · Reviewer_RwCK · 2025-10-30

**Soundness:** 3
**Presentation:** 3
**Contribution:** 3
**Rating:** 8
**Confidence:** 5

**Summary:**

The paper tackles the challenge of automatically evaluating textual creativity in large language models. Current creativity assessment rely heavily on human judgement, which is subjective, inconsistent, and expensive.
The author propose a new evaluation framework which is a context-aware, pairwise comparison method that judges creativity between two responses to the same prompt. The author propose a new dataset containing over 1 million instruction-response pairs across 87 domains, mixing human and synthetic data. Over this, the author train a new LLM-based creativity evaluation on the dataset.

**Strengths:**

1. The overall idea is clear and well-motivated. Focusing on fine-grained text-level creativity rather than broad model-level creativity makes the work much more practically valuable.
2. The authors explicitly explore the challenge of human annotation inconsistency through empirical analysis, which shows solid motivation and thoughtful experimental design.
3. The two key assumptions used for weak supervision are both empirically validated, and the paper carefully considers position bias in pairwise evaluation that is easily to be ignored.
4. The experimental setup is comprehensive and convincing. The baselines cover a wide range of categories—traditional metrics, general LLMs, and fine-tuned evaluators. The study includes analyses of consistency, ablation on data composition, OOD generalization, and even the ability to enhance creativity through DPO training. Altogether, this gives the results strong credibility.

**Weaknesses:**

1. There’s no discussion of whether the model maintains interpretative correctness—in other words, does it still “understand” why a response is creative under its defined criteria?
2. The OOD evaluation is relatively small in scale, which limits how strongly the generalization claim can be made.
3. Some references, such as Zhao (2024), have already been officially published, but the citations still point to arXiv versions. The bibliography needs to be updated.

**Questions:**

1. The CreataSet-Base domain distribution seems quite unbalanced. Is creativity really isotropic across categories? Some subdomains (like business writing vs. poetry) might not be equally suited for creativity evaluation. A more fine-grained per-domain analysis would help clarify this.
2. Were the choices of Qwen2.5-14B-Instruct and MiniCPM-2B-SFT arbitrary? Why were these two particular models selected for response generation and augmentation?
3. The pairwise comparison framework works well for controlled benchmarking, but its practical usability might be limited. Do you think it could be extended to continuous or scale-based scoring while preserving consistency?
4. It would be interesting to test CrEval on more classical creativity benchmarks—like Alternative Uses or other divergent-thinking tasks—where the “good” and “bad” responses are clearer.
5. In real-world creative contexts, can CrEval replace or complement the Consensual Assessment Technique (CAT)?

---

> ### Author Response · Authors · 2025-11-20
> **Response to Reviewer RwCK [1/2]**
>
> Thank you for the recognition of our work and your insightful comments. We will further add the details we missed as mentioned and revise some non-standard points (for example, updating bibliography citations as mentioned in weakness 3). Below, we address your concerns.
>
> ### **1. Model interpretability (Weakness 1)**
>
> We agree that it is essential to ensure the model not only correctly judges creative responses but also maintains a coherent understanding of why those responses are creative. To showcase this aspect of value, we conducted an initial analysis on the test set (3K+ pairs) as shown below. For every pairwise comparison, we use an LLM (i.e., DeepSeek-V3.2) to identify which creative attributes were associated with the response CrEval judged as more creative. We then aggregated these attributes. The distribution reveals that CrEval is not relying on superficial artifacts but consistently attends to semantic, stylistic, and structural patterns that align with widely accepted dimensions of creativity. The most frequent attributes include:
>
> | Category | Ratio |  | Category | Ratio |
> |:-:|:-:|-|:-:|:-:|
> | Unique Imagery | ≈21% |  | Concrete Details | ≈5% |
> | Vivid Metaphor | ≈19% |  | Rich Visuals | ≈5% |
> | Unconventional Expression | ≈12% |  | Imaginative elaboration | ≈4% |
> | Sincere Emotion | ≈10% |  | Distinctive Layers | ≈3% |
> | Profound Symbolism | ≈6% |  | Precise word choice | ≈2% |
>
> These proportions indicate that CrEval internalizes a multidimensional notion of creativity: it prioritizes originality in imagery and figurative expression, while also recognizing emotional depth, semantic richness, and detailed, imaginative construction. This analysis triggers some future work to systematically categorize these emerging patterns and quantitatively evaluate their correlation with human preference judgments. We believe this will bridge the gap between implicit preference learning and explicit, interpretable creativity analysis, making both the dataset and model valuable resources for fundamental creativity research.
>
>
> ### **2. Limited evaluation scope (Weakness 2, Question 4)**
>
> We would like to clarify that the relatively small scale of our OOD evaluation is primarily due to the scarcity of datasets that satisfy two key criteria: (1) being creativity-oriented, and (2) containing human-annotated pairwise preferences for meta-evaluation. To address this and strengthen our generalization claims, we have followed the reviewer's excellent suggestion and conducted a new experiment on an Alternative Uses Task (AUT) dataset [1], which contains 541 human-annotated preference pairs.
>
> The results in the table below show that CrEval not only achieves the best performance among models of similar scale (~7B) but also surpasses much larger models like GPT-4o, Claude-3.5-Haiku, and Gemini-2.5-Pro. Crucially, our training data contained no samples similar to AUT tasks. This strongly demonstrates CrEval's robust generalization ability and its capacity to capture a fundamental, task-agnostic notion of creativity that aligns with human judgment.
>
> | Methods | Agreement | F1 | Consistency | | Methods | Agreement | F1 | Consistency |
> |:---------|:-----------:|:----:|:----:|-|:---------|:-----------:|:----:|:----:|
> | *Frontier* | | | | | *7B Scale* | | | |
> | o3 | 83.55 | 86.51 | 93.53 |  | Gemma-2-9B-it | 68.39 | 89.06 | 74.12 |
> | o1 | 82.07 | 86.23 | 90.39 |  | LLaMA3.1-8B-Inst. | 55.27 | 83.32 | 65.62 |
> | GPT-4o | 62.29 | 75.42 | 71.90 |  | PandaLM-7B | 62.66 | 78.00 | 69.32 |
> | GPT-3.5 | 60.87 | 75.22 | 69.57 |  | Prometheus-7B  | 4.81 | 4.81 | - |
> | DeepSeek-R1 | 61.74 | 75.42 | 71.35 |  | AUTO-J | 62.66 | 73.79 | 74.86 |
> | DeepSeek-V3 | 71.35 | 81.05 | 79.85 |  | WritingBench-Critic  | 69.13 | 69.13 | - |
> | Claude-3.5-Sonnet | 84.66 | 89.83 | 88.91 |  | Qwen2.5-7B-Inst. | 61.37 | 84.17 | 73.20 |
> | Claude-3.5-Haiku | 73.94 | 81.33 | 83.73 |  | CrEval-7B (ours) | **78.37** | **92.46** | **84.29** |
> | Gemini-2.5-Pro | 57.49 | 67.74 | 79.48 |  |
> | Gemini-2.5-Flash | 65.62 | 79.94 | 70.24 |  |
> | G-Eval (GPT-4o) | 66.54 | 76.80 | 77.26 |  |
> | G-Eval (GPT-3.5) | 15.16 | 56.01 | 17.01 |  |
>
> > [1] Sun, Luning, et al. "A New Dataset and Method for Creativity Assessment Using the Alternate Uses Task." BenchCouncil International Symposium on Intelligent Computers, Algorithms, and Applications. Singapore: Springer Nature Singapore, 2023.

---

> ### Author Response · Authors · 2025-11-20
> **Response to Reviewer RwCK [2/2]**
>
> ### **3. Unbalanced creativity domains (Question 1)**
>
> This is a valuable observation. We agree that creativity manifests differently across domains, and our current work intentionally focuses on creativity-dense contexts (e.g., poetry, humor) where its expression is most overt, as noted in Line 198.
>
> To ensure that even domains like business writing in our dataset are conducive to creativity evaluation, we implemented a key filtering step: both instructions and responses were filtered by GPT-4o-mini for their creative potential. Samples consistently receiving low creativity scores were discarded. This process helps ensure that the retained data across all subdomains possesses a sufficiently high potential for creative expression.
>
> Building on this foundation, we fully agree that a fine-grained, per-domain analysis is an essential next step. Systematically investigating how to best characterize and measure creativity in different fields is an excellent direction for future work, which will significantly enhance the applicability and precision of our framework.
>
>
> ### **4. Model Selection Justification (Question 2)**
>
> Our choice of Qwen2.5-14B-Instruct and MiniCPM-2B-SFT was deliberate and served two key methodological purposes. First, the significant scale difference (14B vs. 2B) between these models was essential for testing our first hypothesis that models of varying capabilities exhibit different levels of creative potential. Second, and equally important, we intentionally selected models from different sources with different architectural families (Qwen vs. MiniCPM) rather than using different sizes of the same model. This approach ensures greater distributional diversity in the generated responses, effectively mitigating the risk of data homogenization and creating a more robust and representative evaluation benchmark.
>
>
> ### **5. Extension to continuous scoring (Question 3)**
>
> This is an insightful question regarding the practical application of our framework. While pairwise comparison is ideal for such controlled benchmarking, it can indeed be extended to continuous scoring while maintaining consistency through several approaches. The most straightforward method would implement the Elo rating system, where CrEval compares new responses against calibrated benchmarks to generate stable, relative scores. More directly, the token probabilities from CrEval's preference judgments can serve as continuous confidence scores, preserving the relative nature of pairwise assessment while providing scalar outputs.
>
> Notably, these approaches make CrEval naturally suitable as a reward model in model training, such as reinforcement learning, where it could provide continuous preference signals to guide creative generation. The pairwise framework thus can adapt to these scoring methods, and we consider developing such reliable continuous assessment mechanisms based on CrEval to be a promising direction for future work.
>
>
> ### **6. CrEval vs. CAT (Question 5)**
>
> CrEval is not designed to replace the Consensual Assessment Technique (CAT), but rather to complement it by offering a scalable and automated alternative for specific use cases. While CAT remains a standard for in-depth, expert-driven evaluation in creative domains, CrEval provides an efficient and consistent method for high-volume evaluation, such as during iterative model training or large-scale creativity screening. Thus, the two approaches can be used synergistically: CrEval can rapidly filter or rank large sets of responses, while CAT can be applied for final validation or in-depth analysis of top candidates. This combination balances scalability with depth, making creative evaluation more practical without sacrificing rigor.

---

> > ### Comment · Reviewer_RwCK · 2025-11-27
> >
> > Thanks for your thoughtful response. I will keep my score

---

> > > ### Author Response · Authors · 2025-11-27
> > > **Thanks**
> > >
> > > Thanks for your feedback and support. Please do not hesitate to reach out if you have any further questions. We remain open to discussion at any time.

---

### Official Review · Reviewer_s6ik · 2025-10-31

**Soundness:** 3
**Presentation:** 3
**Contribution:** 3
**Rating:** 6
**Confidence:** 3

**Summary:**

This paper focuses on evaluating the creativity of LLM-generated text. Specifically, it identifies two key challenges in evaluating creativity: 1) ensuring consistency in creative annotation for human experts; and 2) training a large-scale model for evaluating creativity given the scarcity of evaluation corpora. To address these challenges, this paper proposes CreataSet and CrEval, a large-scale pairwise comparison framework for cross-domain text creativity evaluation and a logic learning model (LLM)-based evaluator, respectively. Testing on various baseline models validates the effectiveness of CreataSet in evaluating creativity.

**Strengths:**

1) Large-scale and multi-domain dataset. CreataSet includes 100K+ human-level and 1M+ synthetic creative instruction-response pairs across 87 domains, which is promising in providing a scalable fundation for studying creative generation and evaluation.
2) Improved human label protocol. The proposed context-aware pairwise comparison protocol improves inter-annotator consistency (evaluated by ICC).
3) Comprehensive experiments. Multiple metics are applied for providing an through evaluation, such as F1score, Kappa score and Agreement rate.

**Weaknesses:**

1. The rules for quantifying creativity are not differentiated across different domains.For example, creativity is manifested differently in poetry and scientific writing. Future work should further differentiate the measurement of creativity for each domain.
2. Insufficient example/failure case analysis. The paper presents overall statistics but does not systematically list typical examples of discrepancies between CrEval and human behavior.
3. Insufficient generalization analysis. The paper lacks an assessment of the enhancement effect of CreataSet on diverse baseline models.

**Questions:**

Please refer to Weaknesses.

---

> ### Author Response · Authors · 2025-11-13
> **A Polite Inquiry Regarding Your Feedback**
>
> Dear Reviewer s6ik,
>
> Thank you for your valuable comment. We would like to make sure we fully understand your concern regarding the Weaknesses #3.
> Our current experiments include comparisons across multiple baseline models and methods, and Appendix Table 5 reports results using different backbone architectures. Could you please clarify whether you are suggesting that we (a) evaluate CreataSet on a broader range of backbone models, or (b) include additional baseline methods for comparison? This clarification will help us strengthen the revision in the most relevant way. Thank you very much!
>
> The Authors

---

> ### Author Response · Authors · 2025-11-20
> **Response to Reviewer s6ik**
>
> Thanks for the insightful comments. We address your concerns below.
>
> ### **1. Creativity not differentiated across different domains (Weakness 1)**
>
> Good suggestions! Our current work intentionally focuses on creativity-dense domains (e.g., poem & humor), as outlined in Line 198, where the expression of creativity is more overt and central to the instructions. This initial scope allows us to establish a robust general framework for assessing creativity in contexts where it is a primary objective, as opposed to more constrained domains like scientific writing or straightforward news reporting.
>
> Building upon this foundation, we agree with the reviewer that future work should and will delve deeper into the differences of creativity across different fields. Differentiating the measurement metrics for specific domains is an excellent direction for extending the applicability and precision of our framework to a broader and more diverse range of scenarios.
>
>
> ### **2. Insufficient failure case analysis (Weakness 2)**
>
> We agree that analyzing failure cases is crucial. While Appendix A.4 (Table 8) provides successful qualitative examples, we have included additional failure case analyses below to offer a more balanced and in-depth examination of the discrepancies between CrEval and human judgments.
> ```
> Instruction: Please describe the distance between people in one sentence.
> Response 1: A wall; a bridge; an embrace.
> Response 2: Amid the bustling warmth of human life, only a gentle word lies between heart and heart.
> Ground Truth: Response 1
> CrEval: After careful consideration, I think Response 2 is more creative.
>
> Instruction: Why do people always seem to lose one sock?
> Response 1: In the sock world's ballroom, solo dancers always lose their way.
> Response 2: If both went missing, you wouldn't even notice.
> Ground Truth: Response 2
> CrEval: After careful consideration, I think Response 1 is more creative.
> ```
> These failure cases suggest that when the creativity levels between two responses are very close, CrEval may encounter challenges in accurately identifying subtle semantic preferences. In both examples, while the selected responses demonstrate clear creative merit, the competing alternatives also contain distinctive creative elements, such as metaphorical depth or unexpected perspectives, that appear to resonate differently with human judges. We will further investigate these failure patterns in future work to better understand the nuances in creative evaluation.
>
>
>
> ### **3. Lack of diverse baselines analysis (Weakness 3)**
>
> As shown in Table 2 and further elaborated in Appendix A.3.3 (Table 5), we have already included comparisons across multiple baseline models and various backbones. To further address the reviewer's concerns regarding generalization, we have expanded these experiments. Specifically, we additionally evaluated the performance using DeepSeek-R1-0528-Qwen3-8B (DS-Qwen3-8B) and Qwen3-8B as backbone models. The results below compare the original backbones against their fine-tuned versions on our dataset.
>
> | Models | F1 | Kappa | Agreement |
> |:-|:-|:-|:-|
> | Baichuan2-7B-Chat | 0.474 | 0.167 | 0.265 |
> | Baichuan2-7B-Chat-finetuned | 0.721 (+52.1%) | 0.588 (+252%) | 0.741 (+180%) |
> | Llama3.1-8B-Chat | 0.628 | 0.429 | 0.550 |
> | Llama3.1-8B-Chat-finetuned | 0.729 (+16.1%) | 0.590 (+37.5%) | 0.738 (+34.2%) |
> | DS-Qwen3-8B | 0.647 | 0.430 | 0.526 |
> | DS-Qwen3-8B-finetuned | 0.742 (+14.7%) | 0.603 (+40.2%) | 0.758 (+44.1%) |
> | Qwen3-8B | 0.674 | 0.505 | 0.583 |
> | Qwen3-8B-finetuned | 0.741 (+9.9%) | 0.604 (+19.6%) | 0.737 (+26.4%) |
>
> As the new results demonstrate, both weak (like Baichuan2) and strong (like Qwen3) backbone models achieve significant performance gains after training on our dataset. This consistently significant improvement across diverse architectures strongly underscores the high quality and generalizability of our proposed dataset. Furthermore, it confirms that the performance of our pairwise comparison-based creative evaluation method can be robustly enhanced.

---

> > ### Comment · Reviewer_s6ik · 2025-11-28
> >
> > Thanks for the author's reply and additional experimental results. I’ll keep my score.

---

### Official Review · Reviewer_AWwk · 2025-11-01

**Soundness:** 3
**Presentation:** 2
**Contribution:** 2
**Rating:** 4
**Confidence:** 3

**Summary:**

This paper introduces CreateSet, a dataset with evaluation pipeline including a performant evaluator—CrEval. The author enriched the responses to existing prompts by generating more responses (and formed CreateaSet-Ext). The work annotated over 3,000 samples, with more using weak labels. The author extensively studied if the dataset helped create a more power evaluator.

**Strengths:**

- A reasonably well-curated dataset with good mix of topics for coverage.
- Good amount of human labels, and put into good use in training evaluator models with good comparison with other metrics and very extensive model lineup. I appreciate the authors showing many proprietary results.
- Evaluation of the trained evaluator is complete and convincing.

**Weaknesses:**

- The definition of creativity, which is subjective, should be detailed better in this work. This is a key bottleneck of this work's quality and rigor.
- CrEval are comparison pairs of creativity. However, there might be some value to an absolute scale of creativity, especially if we want to rank model responses quickly. Motivation here is less clear.
- This work suffers from a few overstatements:
    - The paper prides itself over context awareness (i.e., showing a prompt when evaluating responses for creativity), including using an entire Fig 1 to emphasize. But the authors fail to explicitly demonstrate if this is common.
    - "87 domains" is a stretch. "Domain" is defined too loosely across the paper.
- While I believe performance parity with o3 is not so important given the model scale difference, the authors could use more motivation on why a smaller model as an evaluator is important when a large model can do better.

I found these weaknesses not fundamental and am happy to update my views during the coming discussion period.

**Questions:**

- Line 43, How is problem-solving a single domain?
- It is important to break down the language used in the dataset. Is this dataset 100% (Simplified) Chinese?
- Line 47 "most methods evaluate creativity at the model or subject level rather than at the level of individual responses" Please better support this claim.
- How are deepseek models "Proprietary LLMs"?
- How are annotators compensated?

---

> ### Author Response · Authors · 2025-11-20
> **Response to Reviewer AWwk [1/3]**
>
> We thank the reviewer for the constructive feedback. Below, we address the concerns raised.
>
> ### **1. Subjectiveness of creativity definition (Weakness 1)**
>
> **Subjectivity is inevitable but controllable.** We understand the concerns. While creativity is inherently subjective, we wish to clarify that the subjectivity of creativity is inevitable but controllable through our rigorous evaluation design. Although creativity involves personal and cultural variation, psychological studies [1,2] show that people often converge in recognizing creative content. For instance, more creative ideas, such as the novel Harry Potter, typically receive higher engagement (e.g., more likes). Our goal is not to eliminate subjectivity but to model shared human judgment in a reproducible and structured way using pairwise comparisons and consensus-driven aggregation.
>
> This is supported by strong inter-annotator agreement (ICC = 0.75) among our 30 annotators from diverse backgrounds, a level typically considered highly consistent. While it is difficult to be empirically representative of the global majority, this high consensus among a substantial and diverse group marks a significant advance over prior work relying on only 2 [3] or 10 [4] annotators. We thank the reviewer for this suggestion and will elaborate more on these points in the revised manuscript.
>
> > [1] Barbot, Baptiste, Richard W. Hass, and Roni Reiter-Palmon. "Creativity assessment in psychological research:(Re) setting the standards." Psychology of Aesthetics, Creativity, and the Arts 13.2 (2019): 233.
> >
> > [2] Parkhurst, Howard B. "Confusion, lack of consensus, and the definition of creativity as a construct." The Journal of Creative Behavior 33.1 (1999): 1-21.
> >
> > [3] Stevenson, Claire E., et al. "Putting GPT-3's Creativity to the (Alternative Uses) Test." ICCC. 2022.
> >
> > [4] Chakrabarty, Tuhin, et al. "Art or artifice? large language models and the false promise of creativity." Proceedings of the 2024 CHI Conference on Human Factors in Computing Systems. 2024.
>
>
> ### **2. Comparison with the absolute scale of creativity (Weakness 2)**
>
> We wish to clarify that we do not intend to dismiss the value of absolute score evaluation, but rather to emphasize that the pairwise comparison approach offers several distinct advantages.
>
> **Absolute creativity scales are difficult to define and apply.** In practice, defining a universal absolute scale for creativity is difficult: annotators find it hard to define what 1 to 5 means across samples and keep consistent standards/thresholds across people. Depending on the context of the instruction, a response scoring 3 could be deemed creative, whereas another instruction might require a 5 score for its response to constitute a creative answer, making "high score = high creativity" an unreliable standard. In our pilot experiments, the ratings of 3 annotators on 50 data groups (each with 5 responses) demonstrate that they produced similar relative rankings across responses but diverged substantially in absolute scores, with differences up to 1.02 points. This level of inconsistency further reduces the usefulness of absolute scoring for large-scale alignment.
>
> **Pairwise comparison aligns directly with how LLMs are trained.** We frame CrEval as a pairwise task due to its direct applicability to various model training algorithms, including DPO, reward model training, etc. These methods require a reliable, scalable, and implicit understanding of preference. Given the above challenges, absolute scale methods like prompt-based LLM scoring[1,2], often struggle to provide fine-grained, relative assessments needed for model alignment. This is partly because the definition of an absolute score can be ambiguous and prone to individual interpretation, leading to inconsistency. As a result, pairs derived from absolute scores tend to be less reliable than those constructed directly through pairwise comparison, which provides clearer and more consistent training signals. If we want to quickly rank model responses using pairwise comparisons, we can leverage the response from any model as a reference. Ranking can then be efficiently achieved based on win rate, incurring minimal computational overhead. These advantages above motivate our design choice for CrEval.
>
> > [1] Summers-Stay, Douglas, Clare R. Voss, and Stephanie M. Lukin. "Brainstorm, then select: a generative language model improves its creativity score." The AAAI-23 Workshop on Creative AI Across Modalities. 2023.
> >
> > [2] Zhao, Yunpu, et al. "Assessing and Understanding Creativity in Large Language Models." Machine Intelligence Research 22.3 (2025): 417-436.

---

> ### Author Response · Authors · 2025-11-20
> **Response to Reviewer AWwk [2/3]**
>
> ### **3. Overstatements on context awareness and diverse domains (Weakness 3)**
>
> **(1) Discussion on context awareness**
>
> We agree that the term "context-awareness" requires clarification, and we will further discuss this, focusing on two key aspects:
>
> **a) Methodological Necessity & Data Construction**: Context-aware evaluation serves as a methodological precondition of our work, rigorously embedded into the fabric of our dataset. Unlike standalone text collections, our instances are explicitly constructed as (instruction, response) pairs. As detailed in Fig. 2 & Fig. 3, we meticulously filtered and transformed data sources to ensure every creative evaluation is grounded in a specific context. This is crucial because a response's creativity is inherently relative to the instruction that elicited it. Even when one thinks evaluating a standalone text is possible, it often implicitly relies on a hypothetical, shared context, whereas our method makes this context explicit and controlled. For instance, when evaluating a poem, one must first consider its theme (e.g., homesickness) and then compare it to other poems on the same theme to identify novelty, rather than making cross-context comparisons.
>
> **b) Conceptual Distinction from Prior Work**: While some prior works like [1] use query-dependent evaluation (often for quality or style in writing tasks) and datasets like [2] contain instructions, they do not explicitly position or investigate context as the fundamental factors for creativity evaluation. Our work moves beyond simply having a context to arguing for its indispensability. We demonstrate that true creativity is a relational property between a response and its prompt, not an intrinsic property of text. We hope this conceptual framing through our benchmarking can advance the methodology of computational creativity evaluation.
>
> **(2) The definition of "domains"**
>
> The term "domain" in our work refers to a thematic category identified through a systematic, two-stage classification process, designed specifically to demonstrate the diversity of our dataset beyond a single style or field. Our methodology for domain extraction follows established practices in prior works [2,3,4]. Starting from a manually curated seed taxonomy, we employed GPT-4o-mini to classify each data sample into a fine-grained category, yielding 87 distinct subdomains. They were then aggregated by the model into broader, semantically coherent ones, resulting in the final set of 17 core domains. Manual inspection confirmed that the resulting categories show minimal overlap and maintain high conceptual clarity. This automated, two-stage approach effectively served our goal of demonstrating data diversity rather than adopting a prohibitively expensive manual taxonomy. We will clarify this methodology and the intended meaning of "domain" in a revision to prevent any perception of overstatement.
>
>
> > [1] Wu, Yuning, et al. "Writingbench: A comprehensive benchmark for generative writing." arXiv preprint arXiv:2503.05244 (2025).
> >
> > [2] Tian, Yufei, et al. "MacGyver: Are Large Language Models Creative Problem Solvers?." Proceedings of the 2024 Conference of the North American Chapter of the Association for Computational Linguistics: Human Language Technologies (Volume 1: Long Papers). 2024.
> >
> > [3] Wang, Yizhong, et al. "Self-instruct: Aligning language models with self-generated instructions." Proceedings of the 61st annual meeting of the association for computational linguistics (volume 1: long papers). 2023.
> >
> > [4] Jin, Chuhao, et al. "Persuading across diverse domains: a dataset and persuasion large language model." Proceedings of the 62nd Annual Meeting of the Association for Computational Linguistics (Volume 1: Long Papers). 2024.
>
>
>
> ### **4. Importance of smaller evaluators (Weakness 4)**
>
> We agree that absolute performance parity with a much larger model like o3 is not the primary goal. Instead, the key motivation for developing a smaller, specialized evaluator like CrEval lies in its practical advantages for the model development lifecycle. It offers significantly lower inference cost and computational overhead, enabling rapid, frequent, and cost-effective evaluation during iterative training cycles (e.g., for reward models or during DPO). Furthermore, its smaller size allows for easy offline deployment, facilitating local and private evaluation of creativity, which is crucial for many research and industrial applications where data privacy or API latency are concerns.

---

> ### Author Response · Authors · 2025-11-20
> **Response to Reviewer AWwk [3/3]**
>
> ### **5. Questions**
>
> **(Question 1): Line 43, How is problem-solving a single domain?**
>
> We apologize for the lack of clarity in our original statement. We intended to indicate that the "problem-solving" tasks referenced (e.g., using tools to solve a specific problem [1] or finding a word that meets certain criteria [2]) are often confined to a single, constrained task format. This is different from open-domain instruction following, where the potential domains and response styles are vastly broader. When classified using our same methodology, the dataset in [1], for instance, spans only 14 domains, confirming its relatively narrow thematic scope compared to the diversity we aim for. We will revise the wording in the manuscript to accurately reflect this distinction and thank the reviewer for highlighting this ambiguity.
>
> > [1] Tian, Yufei, et al. "MacGyver: Are Large Language Models Creative Problem Solvers?." Proceedings of the 2024 Conference of the North American Chapter of the Association for Computational Linguistics: Human Language Technologies (Volume 1: Long Papers). 2024.
> >
> > [2] Alavi Naeini, Saeid, et al. "Large language models are fixated by red herrings: Exploring creative problem solving and einstellung effect using the only connect wall dataset." Advances in Neural Information Processing Systems 36 (2023): 5631-5652.
>
>
> **(Question 2): It is important to break down the language used in the dataset. Is this dataset 100% (Simplified) Chinese?**
>
> We thank the reviewer for raising this point. The dataset is predominantly in Simplified Chinese, and approximately 8% of samples contain mixed Chinese-English content. We will clarify this in the revised manuscript. Our focus on a single language stems from the deeply contextual nature of creativity, which is highly subject to cultural and linguistic context. Moreover, it allowed us to recruit qualified assessors and maintain consistent annotation quality. While our work validates the methodology in Chinese, the framework is language-agnostic and can be easily extended to other languages in future work.
>
>
> **(Question 3): Line 47 "most methods evaluate creativity at the model or subject level rather than at the level of individual responses". Please better support this claim.**
>
> Existing LLM creativity evaluations largely inherit frameworks from psychology, where creativity is assessed at the subject level using specific creativity tests [1,2] such as TTCT (line 104). These methods measure how a person/model performs on a predefined set of creativity tasks, producing model- or subject-level judgments rather than evaluating the creativity of arbitrary responses. Humans, however, can judge the creativity of any two pieces of work even outside such constrained tasks. Our work follows this broader perspective: we evaluate creativity at the level of individual responses across open-domain instructions, rather than limiting evaluation to performance on specific tasks. This distinction underpins the motivation for our approach.
>
> > [1] Mednick, Martha T., and Sharon Halpern. "Remote associates test." Psychological Review (1968).
> >
> > [2] Torrance, E. Paul. "Torrance tests of creative thinking." Educational and psychological measurement (1966).
>
>
>
> **(Question 4): How are deepseek models "Proprietary LLMs"?**
>
> Thank you for the correction. We will replace "Proprietary LLMs" with a more precise description, such as "Proprietary or Frontier LLMs" in the revised manuscript.
>
>
> **(Question 5): How are annotators compensated?**
>
> The human annotation was conducted through a professional data annotation company. All annotators were well-educated and were hired as temporary research assistants, compensated at a rate of 50 RMB per hour, which is competitive for skilled academic work in the local context.

---

### Official Review · Reviewer_Re9M · 2025-11-01

**Soundness:** 2
**Presentation:** 3
**Contribution:** 2
**Rating:** 4
**Confidence:** 5

**Summary:**

This paper tackles the problem of evaluating text creativity by building a large-scale dataset (CreataSet and its extended version, CreataSet-Ext) and training a pairwise creativity evaluator CrEval. Each instruction is paired with multiple responses generated by different models under both ordinary and creativity-oriented prompts, with human-annotated labels serving as ground truth. The paper finds that CrEval aligns well with human judgments, generalizes to unseen domains, and can further help enhance model creativity. The results show that creativity-oriented prompts and stronger models tend to produce more creative responses, and data composition meaningfully affects CrEval’s performance.

**Strengths:**

The experiments are fine. CrEval consistently outperforms strong baselines including large proprietary models across proposed metrics. The paper includes appropriate ablations, along with OOD tests on external datasets. The authors further show CrEval can be used to improve model creativity

**Weaknesses:**

-  The paper mentions constructing tuples (I, R1, ..., Rk) but does not specify the exact value of k used in experiments. How many responses are generated and used per instruction? Does this vary across data sources?
- In constructing CreataSet-Ext, they prompt two models to generate more responses for augmenting each instruction. But there is no testing of whether these k responses actually exhibit meaningful diversity. If these responses are similar to each other, it could limit the model’s ability to learn fine-grained distinctions.
- At the beginning of the paper, they define creativity as “ideas or artifacts that are new, surprising and valuable”, which I personally appreciate. While novelty and surprise are well-measured, the “valuable” aspect is not explicitly evaluated. Aside from gpt-4o-mini filtering, there is no systematic check for helpfulness. A response can be novel but incoherent or unhelpful, which weakens the notion of genuine creativity.
-  The two core assumptions are validated with only 50 samples each with three annotators. This seems insufficient given the scale of the dataset (1M+ samples).
- The paper treats creativity evaluation as a pairwise comparison task without deeply analyzing what constitutes creativity or providing any meaningful understanding. What specific features, semantic patterns, or structural elements does CrEval learn to recognize as creative?
- Personally the task are a bit strange and adhoc. I would not treat them as something that requires creativity
-Data contamination could inflate the performance of models being evaluated, especially for proprietary models whose training data composition is not fully disclosed.
- I am not convinced LLM eval is a surrogate for human eval. Are your 18 humans experts ?? Without knowing much we cant conclude anything

**Questions:**

NA

---

> ### Author Response · Authors · 2025-11-13
> **A Polite Inquiry Regarding Your Feedback**
>
> Dear Reviewer Re9M,
>
> Thank you for your insightful feedback. To ensure we address your concerns precisely, we would like to kindly seek clarification on two points:
>
> > Personally the task are a bit strange and adhoc. I would not treat them as something that requires creativity -Data contamination could inflate the performance of models being evaluated, especially for proprietary models whose training data composition is not fully disclosed.
>
> - Rgarding your above comment, could you please let us know whether this was a single combined question or two separate points? (We are asking because the dash might reflect Markdown list syntax rather than indicate a conceptual link.) If it is a single issue, we would greatly appreciate any additional details you could share to help us better understand your point.
>
> > I am not convinced LLM eval is a surrogate for human eval. Are your 18 humans experts ?? Without knowing much we cant conclude anything
>
> - With respect to your point on human evaluation, we would like to note that our study involved 30 human annotators (not 18). Could you please clarify whether your concern mainly pertains to (a) the expertise level of the annotators, (b) the relationship between the LLM-based and human evaluations, or (c) both?
>
> Your clarification would help us respond to your comments in a more precise and constructive manner. Thank you!
>
> The Authors

---

> ### Author Response · Authors · 2025-11-20
> **Response to Reviewer Re9M [1/3]**
>
> Thank you for the valuable comments. Below, we address your concerns and offer further discussion.
>
> ### **1. Insufficient data construction details and diversity validation (Weakness 1 & 2)**
>
> **(1) Value of k:**
>
> We used k = 5 responses per instruction in our experiments. This setting is shared across all data sources. Each instruction is paired with one human-level response, plus four model-generated responses: two models (a stronger vs. a weaker one) × two prompt types (creative vs. ordinary), based on our two hypotheses. We did not increase k further because the pairwise construction already produces a large number of training pairs, and our dataset contains over 110k instructions, which ensures sufficient overall diversity.
>
> **(2) Diversity of generated responses:**
>
> Following the reviewer's suggestion, we evaluated whether the k responses meaningfully differ from one another. Using the Qwen3-Embedding-8B model, we computed semantic distances (cosine similarities) for the training pairs. The distances span 0.19–0.94, with a median of 0.64, showing that the responses vary across both fine-grained and coarse semantic differences. We also conducted a human diversity assessment: 100 randomly sampled groups (k=5 responses each) of responses were rated by 3 annotators on a 1–5 scale. The diversity scores fall within 1–5, with an average score of 3.84. This confirms that the responses are not clustered around a narrow semantic band. These results indicate that our data exhibit substantial and meaningful diversity, which supports learning both subtle distinctions (when responses are semantically close) and broader conceptual differences (when they diverge).
>
>
> ### **2. Lack of systematic check of "valuable" (Weakness 3)**
>
> "Valuable" is explicitly enforced in our filtering criteria: We agree with the reviewer that novelty without any value is not considered creative. The "valuable" aspect of creativity is required for both our LLM filtering and human annotation. In the LLM-based filtering prompt (Table 11 in the Appendix), responses must be "novel and meaningful," and "excessive repetition and commonplace expressions should be assigned low scores". Our human-annotation guideline likewise specifies that creative but unhelpful or ill-formed responses are not considered creative ("consider both the novelty/originality and the relevance", "If the reply does not correspond to the instruction, ..., marked as -1 point."). These methods ensure that the dataset only retains outputs meeting the full definition of creativity. Below, we show several incoherent or low-creativity cases that were filtered out.
>
> ```
> Instruction: How to collect many stars?
> Response (filtered out): I want to listen to this side.
>
> Instruction: Common things for people who always have headphones.
> Response (filtered out): Often listen to music with headphones.
>
> Instruction: Write a song in the style of a bad girl.
> Response (filtered out): Bad girl, bad girl, bad girl, bad girl, bad girl, bad girl, bad girl, bad girl, ......
> ```
>
> ### **3. Insufficient statistical validation of the assumptions (Weakness 4)**
>
> Sorry for the lack of clarity. Our initial study followed the setup in [1], where we manually annotated 50 groups (actually 1 instruction + 5 responses), yielding 350 pairwise samples rather than only 50 samples. To further address the reviewer’s concern, we increased the validation size to 150 groups, resulting in more than 1,000 paired evaluations. We ask 3 annotators to score each response on a scale of 1-4. Under this expanded setting, Assumption 1 reaches an accuracy of 86.6%, and Assumption 2 reaches 81.4%. The enlarged annotation confirms that both assumptions hold robustly, supporting their use in constructing our large-scale dataset.
>
> > [1] Chakrabarty, Tuhin, et al. "Art or artifice? large language models and the false promise of creativity." Proceedings of the 2024 CHI Conference on Human Factors in Computing Systems. 2024.

---

> ### Author Response · Authors · 2025-11-20
> **Response to Reviewer Re9M [2/3]**
>
> ### **4. Lack of insight into learned creativity features (Weakness 5)**
>
> Good suggestions! We treat pairwise comparison as the primary task due to its direct applicability to various model training algorithms, including DPO, reward model training, etc. These methods require a reliable, scalable, and implicit understanding of preference rather than an explicit, human-defined set of creativity features or direct creativity scores without comparison. However, existing works typically rely on manually designed creativity criteria [1,2] or prompt-based LLM scoring [3,4], which often lack robust support for pairwise comparison and struggle to provide the fine-grained, relative assessments needed for model alignment.
>
> > [1] He, Qianyu, et al. "HAUSER: Towards Holistic and Automatic Evaluation of Simile Generation." Proceedings of the 61st Annual Meeting of the Association for Computational Linguistics. 2023.
> >
> > [2] Lu, Ximing, et al. "AI as Humanity’s Salieri: Quantifying Linguistic Creativity of Language Models via Systematic Attribution of Machine Text against Web Text." The Thirteenth International Conference on Learning Representations. 2024.
> >
> > [3] Summers-Stay, Douglas, Clare R. Voss, and Stephanie M. Lukin. "Brainstorm, then select: a generative language model improves its creativity score." The AAAI-23 Workshop on Creative AI Across Modalities. 2023.
> >
> > [4] Zhao, Yunpu, et al. "Assessing and Understanding Creativity in Large Language Models." Machine Intelligence Research 22.3 (2025): 417-436.
>
> Your comments remind us that our dataset provides further value for deeply analyzing the latent factors of creativity. To showcase this aspect of value, we conducted an initial analysis on the test set (3K+ pairs) as shown below. For every pairwise comparison, we use an LLM (i.e., DeepSeek-V3.2) to identify which creative attributes were associated with the response CrEval judged as more creative. We then aggregated these attributes. The distribution reveals that CrEval is not relying on superficial artifacts but consistently attends to semantic, stylistic, and structural patterns that align with widely accepted dimensions of creativity. The most frequent attributes include:
>
> | Category | Ratio |  | Category | Ratio |
> |:-:|:-:|-|:-:|:-:|
> | Unique Imagery | ≈21% |  | Concrete Details | ≈5% |
> | Vivid Metaphor | ≈19% |  | Rich Visuals | ≈5% |
> | Unconventional Expression | ≈12% |  | Imaginative elaboration | ≈4% |
> | Sincere Emotion | ≈10% |  | Distinctive Layers | ≈3% |
> | Profound Symbolism | ≈6% |  | Precise word choice | ≈2% |
>
> These proportions indicate that CrEval internalizes a multidimensional notion of creativity: it prioritizes originality in imagery and figurative expression, while also recognizing emotional depth, semantic richness, and detailed, imaginative construction. This analysis triggers some future work to systematically categorize these emerging patterns and quantitatively evaluate their correlation with human preference judgments. We believe this will bridge the gap between implicit preference learning and explicit, interpretable creativity analysis, making both the dataset and model valuable resources for fundamental creativity research. We will incorporate these discussions in the revised manuscript.
>
>
>
> ### **5. Data contamination concerns (Weakness 6)**
>
> Robustness through Comparative Assessment: Critically, our pairwise evaluation framework is inherently robust to this issue. Since the task requires a relative judgment between two creative responses rather than an absolute score, it minimizes the advantage from merely recognizing familiar content from its training patterns (e.g., contamination) and instead directly probes a model's alignment with nuanced human creativity preferences.
>
> Empirical Validation from Fine-tuning: Our core findings focus on the improvement of open-source models (like Qwen) after training on our dataset. As shown in Table 2, the base models perform poorly, while the same models after training achieve significant performance gains. This clear contrast demonstrates that the capability for comparative creativity is fundamentally attributed to learning from our data, not to pre-existing knowledge or evaluation contamination.

---

> ### Author Response · Authors · 2025-11-20
> **Response to Reviewer Re9M [3/3]**
>
> ### **6. Questioning LLM evaluation validity and human expertise (Weakness 7)**
>
> We clarify that LLM-based evaluation is not intended as a full replacement for human judgment, but rather as a scalable and efficient proxy that provides a valuable reference during rapid model iteration. Embracing such automated methods is a necessary step to advance the field beyond the constraints of slow and costly human-only evaluation, which often hinders the rapid experimentation required for progress in model alignment.
>
> Regarding our human evaluation, we engaged a diverse group of 30 well-educated annotators (aged 21-29, from 18 distinct academic majors) as detailed in Appendix A.7. They underwent rigorous training, and quality control checkpoints were implemented throughout the annotation process. Crucially, since our study targets creativity across diverse domains, as reflected in tasks like humor or open-ended literary creation rather than specialized domains, we intentionally sought to capture the preferences of the general educated public. The diversity in our annotator pool is therefore a strength, ensuring that our human evaluations reflect a broad and representative spectrum of ordinary creative judgment, not just that of domain-specific experts.

---

> > ### Comment · Reviewer_Re9M · 2025-11-27
> > **Thanks**
> >
> > Thanks. Personally I dont think this paper is something I would give a score of 6 to. The core contribution isnt very exciting and it doesnt move the needle much on Creativity research.

---

> ### Author Response · Authors · 2025-11-27
> **Response to Reviewer Re9M**
>
> Dear Reviewer Re9M,
>
> We deeply appreciate your valuable time and thoughtful feedback on our work.
>
> We would like to share a recent observation that resonates with the core idea of our work. A recent perspective [1] from Demis Hassabis on the creativity of future AI highlights that
> > ''make sure that you're not searching that space totally randomly. It would be too big. So you have to have some objective function that you're trying to optimize and hill climb towards and that guides that search''.
>
> This echoes the motivation behind our work, i.e., **to first establish a reliable and learnable objective function for creativity evaluation**, embodied in our CrEval model, which can then guide LLMs toward more creative generation in a scalable and directed manner.
>
> Although defining and measuring creativity is so challenging, we find it encouraging that annotators show high agreement when comparing candidates within the same context. This consensus holds across diverse domains, **suggesting that creative judgments are inherently relative**. People assess a work not against unrelated items, but against existing related works created for the same purpose. Based on this **reliable and domain-agnostic foundation**, our large-scale dataset provides a valuable resource for the community, enabling future studies on creativity.
>
> We do not claim we perfectly defined creativity. Rather, our contribution lies in **breaking the "definition bottleneck" that has long hampered scalable progress in computational creativity**. By providing an actionable methodology to collect contextualized preference data at scale and training an automated evaluator that generalizes robustly, we establish a practical and reproducible pathway for optimizing creativity.
>
> These capabilities are quantitatively evidenced by CrEval's superior alignment with human judgments, its demonstrated ability to enhance model creativity, and the novel insights it enables into creative attributes. The potential of this paradigm has been recognized by other reviewers, who noted that the work is "promising in providing a scalable foundation for studying creative generation and evaluation" (Reviewer `s6ik`) and praised the "clear and well-motivated idea", "comprehensive and convincing experimental setup", and "results strong credibility" (Reviewer `RwCK`).
>
> Thank you once again for considering our work. If you have any further concerns, we always welcome the opportunity for deeper dialogue on this important paradigm.
>
> Best regards,
>
> The Authors
>
> > [1] #475 – Demis Hassabis: Future of AI, Simulating Reality, Physics and Video Games

---

### Author Response · Authors · 2025-11-25
**Summary of Revisions**

We sincerely thank all reviewers for their thorough and constructive feedback, which has been invaluable for improving this work. We have incorporated all reviewer comments into the updated PDF. The main revisions include:

* Revised statements on "problem-solving" and "domain" in the Introduction Section (reviewer `AWwk`).
* Additional clarification on "dataset languages" and "classification methodology" in Section 3.1 (reviewer `AWwk`).
* Expanded analysis of "dataset diversity" in Section 3.2 and Appendix A.3.5 (reviewer `Re9M`).
* Extension of the "validation data" in Section 3.3 (reviewer `Re9M`).
* Further explanation of "hyperparameter" choices in Section 4.1 (reviewer `Re9M`).
* Newly added "O.O.D. experiments on the AUT task" in Section 4.4 (reviewer `RwCK`).
* Additional discussion on "absolute scoring" in Appendix A.2 (reviewer `AWwk`).
* Analysis of the model's "learned creativity features" in Appendix A.4.4 (reviewers `Re9M` & `RwCK`).
* Presentation and analysis of "failure cases" in Appendix A.5 (reviewer `s6ik`).
* Additional details on "annotation information" in Appendix A.8 (reviewer `AWwk`).
* Updated bibliography (reviewer `RwCK`).

For other questions and clarifications, please refer to our detailed responses under each review. We look forward to your consideration of our responses and remain open to discussing any further questions you may have. Thank you again for your time and attention.

---

### Author Response · Authors · 2025-11-27
**[To Reviewers] A Gentle Reminder**

Dear Reviewers,

I hope this message finds you well. As the discussion period enters its final week, we would like to kindly follow up and ensure that we have adequately addressed all of your concerns. Please let us know if there is anything else we can clarify, since your perspective is crucial in helping us improve our work.

Thank you once again for your time and effort in reviewing our paper. We truly appreciate your contributions to this process.

The Authors

---

### Author Response · Authors · 2025-12-02
**Global Response**

We sincerely thank all the reviewers, the ACs/SACs, and the entire PC for their insightful feedback and stewardship throughout the review process. We provide below a concise overview of our core contributions, reviewer concerns, and our responses from the rebuttal to facilitate the AC's decision.

---
### **Our Core Contributions**
- **A scalable, context-aware framework for evaluating creativity**: We design a contextualized pairwise comparison protocol that yields reliable human labels and supports large-scale weakly supervised data generation across diverse domains.

- **A large-scale creativity dataset and a strong LLM evaluator**: We construct **CreataSet**, comprising 100K+ human-level and over 1M synthetic instruction–response pairs spanning 87 creative domains, effectively breaking the long-standing barrier to collecting broad, generalizable creativity data. Building on this corpus, we train **CrEval**, an LLM-based creativity evaluator that achieves substantial improvements over frontier models (e.g., +18.7% over GPT-4o) in alignment with human judgments and demonstrates robust cross-domain generalization.

- **A practical, learnable objective for creativity optimization**: Beyond evaluation, We further demonstrate that CrEval serves as a scalable, learnable objective to guide LLMs toward more creative generation, quantitatively enhancing model creativity. By unifying evaluation and optimization, CrEval opens new avenues for understanding and improving creative attributes in generative AI.

---
### **Reviewer Main Concerns & Our Responses**

- **Data diversity & validation** (`Re9M`, `s6ik`): Added quantitative evidence for diversity (semantic distance/human ratings) and the "valuable" filtering criterion; expanded hypothesis validation to 150 data groups, achieving an accuracy rate of >81%.

- **Subjectivity & Comparison** (`AWwk`): Clarified controlled subjectivity via rigorous pairwise comparison and high inter-annotator agreement (ICC=0.75). Clarified the advantages of paired comparisons over absolute ratings (higher annotation consistency, aligned with preference optimization).

- **More evaluation scope** (`RwCK`, `s6ik`): Conducted additional out-of-distribution (OOD) tests on the AUT dataset, where CrEval surpassed multiple large models such as GPT-4o; expanded experiments with multiple base models, all demonstrating significant improvements.

- **Model interpretability** (`RwCK`, `Re9M`): Post-hoc attribution analysis reveals CrEval aligns with established creativity dimensions (e.g., Unique Imagery ~21%, Vivid Metaphor ~19%), confirming learning of semantic patterns.

We have actively addressed all reviewers' comments and have integrated the revisions from the rebuttal process into the updated version of the paper. All modifications have been highlighted in **blue** for clarity in the new PDF. These updates are also summarized in the `"Summary of Revisions"` below for reference.

---
### **Final Comment**

Our work presents a timely and rigorously validated framework that addresses a critical gap in the systematic evaluation of creativity in LLMs. This paradigm is both scalable and generalizable, establishing a new foundation for studying and optimizing creative generation. The potential of this approach has been recognized by reviewers, who remarked on its "promising in providing a scalable foundation for studying creative generation and evaluation" (Reviewer `s6ik`) and praised its "clear and well-motivated idea," "comprehensive experimental setup," and "strong credibility" (Reviewer `RwCK`).

All substantive reviewer concerns have been carefully addressed with additional experiments, analyses, and clarifications. We trust that the strengthened manuscript now stands as a rigorous and well-supported contribution, ready for the ICLR community through its open dataset, reproducible methodology, and actionable insights into creativity evaluation and enhancement.

---

### Meta-Review · Area_Chair_BbSy · 2026-01-01

**Summary:**

This paper addresses the long-standing challenge of evaluating textual creativity in large language models by proposing a context-aware, pairwise comparison framework that aligns more closely with human judgment than absolute scoring. The authors introduce CreataSet, a large-scale dataset comprising over 100K human-level and more than 1M synthetic instruction–response pairs spanning a wide range of creative domains, and train an LLM-based evaluator, CrEval, on this data. Extensive experiments demonstrate that CrEval substantially outperforms existing automatic metrics and strong proprietary models in alignment with human preferences, generalizes to out-of-distribution creativity benchmarks such as AUT, and can serve as a learnable objective to enhance the creativity of generative models through preference optimization.

The main strengths of the work are the scale and scope of the dataset, the carefully designed pairwise annotation protocol that demonstrably improves inter-annotator consistency, and a comprehensive experimental evaluation covering baselines, ablations, OOD tests, and downstream optimization use cases. The rebuttal substantially strengthened the paper by clarifying data construction details, validating response diversity and key assumptions at a larger scale, addressing concerns about the “valuable” aspect of creativity, adding interpretability analyses, expanding generalization experiments, and correcting overstatements and missing details. Remaining weaknesses are mostly conceptual rather than empirical: creativity is still modeled implicitly rather than with domain-specific criteria, the notion of “domain” remains somewhat coarse, and interpretability analyses are preliminary. However, these limitations are acknowledged and framed as directions for future work, rather than flaws that undermine the core contribution.

Considering the strong empirical contributions and the authors’ thorough and convincing rebuttal, which effectively addressed most substantive concerns, this paper should be accepted.

**Reviewer Concerns:**

The rebuttal successfully resolved the majority of technical and methodological issues raised in the reviews. Questions about dataset construction (e.g., number of responses per instruction, response diversity, and validation scale) were answered with concrete details and new quantitative and human evaluations. Concerns about subjectivity and the definition of creativity were mitigated by clarifying the motivation for pairwise comparison, reporting high inter-annotator agreement, and explaining why relative judgments are more reliable for large-scale alignment. Requests for broader evaluation were addressed through additional OOD experiments, new backbone models, and expanded failure-case analyses. Interpretability concerns were partially addressed through post-hoc analyses, which showed that CrEval attends to semantically meaningful creativity attributes rather than superficial cues. Minor issues such as wording, references, language composition, and annotation details were also clarified.

Some higher-level concerns remain only partially resolved. Creativity is still treated as a largely unified construct across domains, and while the authors justify focusing on creativity-dense tasks, a more principled, domain-specific treatment is left for future work. Interpretability analyses, while helpful, are exploratory and do not yet provide a rigorous theoretical account of what constitutes creativity. Additionally, some reviewers remained unconvinced that the work fundamentally advances the theory of creativity rather than providing a strong engineering and benchmarking contribution. These concerns reflect differences in expectation rather than clear technical deficiencies.

**Reviewer Scores:**

Across the reviews, initial scores clustered around the marginal accept/reject boundary, with one clearly positive evaluation. After the rebuttal and discussion, it is likely that the more skeptical reviews would have modestly increased their scores (e.g., from marginal reject to weak accept or borderline), as many concrete criticisms were addressed and several reviewers explicitly indicated openness to acceptance. Reviews that were already positive maintained their scores. Overall, the distribution would likely shift slightly upward, yielding a consensus consistent with acceptance as a solid but not unanimous contribution.

---

### Decision · Program_Chairs · 2026-01-26

Accept (Poster)